# A unified framework for establishing the universal approximation of transformer-type architectures

**Jingpu Cheng**
Department of Mathematics
National University of Singapore
chengjingpu@u.nus.edu

**Ting Lin**
School of Mathematical Sciences
Peking University
lintingsms@pku.edu.cn

**Zuowei Shen**
Department of Mathematics
National University of Singapore
matzuows@nus.edu.sg

**Qianxiao Li**
Department of Mathematics
Institute for Functional Intelligent Materials
National University of Singapore
qianxiao@nus.edu.sg

## Abstract

We investigate the universal approximation property (UAP) of transformer-type architectures, providing a unified theoretical framework that extends prior results on residual networks to models incorporating attention mechanisms. Our work identifies token distinguishability as a fundamental requirement for UAP and introduces a general sufficient condition that applies to a broad class of architectures. Leveraging an analyticity assumption on the attention layer, we can significantly simplify the verification of this condition, providing a non-constructive approach in establishing UAP for such architectures. We demonstrate the applicability of our framework by proving UAP for transformers with various attention mechanisms, including kernel-based and sparse ones. The corollaries of our results either generalize prior works or establish UAP for architectures not previously covered. Furthermore, our framework offers a principled foundation for designing novel transformer architectures with inherent UAP guarantees, including those with specific functional symmetries. We propose examples to illustrate these insights.

## 1 Introduction

Transformers [66] are a family of deep learning architectures that have achieved remarkable performance in natural language processing [5, 57, 58], computer vision [7, 16], and other fields [36]. Given an input sequence of tokens, a transformer processes it through a deep composition of alternating attention and token-wise feedforward layers. Besides the original softmax attention [66], a variety of different attention mechanisms have been proposed to enhance performance or computational efficiency, such as kernel-based attention [9, 12, 38, 65], sparse attention [3, 40, 76], and attention with low-rank structures [68, 82].

A natural theoretical question is: *What is the expressive power of these architectures?* Previous studies have shown that transformers achieve the universal approximation property (UAP) via architecture-specific constructions, meaning they can approximate any continuous sequence-to-sequence function over compact domains [37, 75, 76] in certain measures. However, these results heavily rely on explicit, architecture-specific constructions, and a unified theoretical framework of deep transformer-type architectures remains elusive. In particular, it is highly desirable to derive a verifiable condition that guarantees UAP of deep transformer-type architectures, independent of specific architectural details

such as the choice of the attention mechanism. Such a framework would allow greater flexibility in design without sacrificing expressivity.

Similar concerns have been addressed for fully connected deep residual networks (ResNets) using insights from control theory and dynamical systems [10, 44, 62, 63]. By interpreting ResNets as control systems, recent studies [10, 44] showed that deep ResNets with Lipschitz nonlinear activation functions possess UAP. However, extending this approach to transformers presents a challenge. Unlike ResNets, transformers apply identical feedforward transformations across tokens, without direct inter-token interactions. Hence, the attention mechanism must effectively capture token dependencies and propagate contextual information throughout the network.

To extend the UAP framework from ResNets to transformers, we model each transformer block through two sequential operations[1]:

$$
\begin{aligned}
X_{t+\frac{1}{2}} &= X_t + \text{Atten}(X_t), \\
X_{t+1} &= X_{t+\frac{1}{2}} + \text{FFN}(X_{t+\frac{1}{2}}),
\end{aligned}
\tag{1}
$$

where $X_t \in \mathbb{R}^{d \times n}$ represents $n$ tokens each of dimension $d$. The feedforward network (FFN) acts independently on each token, while the attention layer ($\text{Atten}$) explicitly models dependencies across tokens. A transformer model is then defined as the composition of such blocks. In many practical architectures, the attention layer is computed by some interactions between tokens, which has a computational complexity up to $\mathcal{O}(n^2 d)$, while the feedforward layer has computation complexity of $\mathcal{O}(nd^2)$. This structure reduces computational complexity from $\mathcal{O}(n^2 d^2)$ (for using network with dense layers) to $\mathcal{O}(nd(n+d))$(can be even lower in many variants) by decomposing approximation tasks into simpler token-wise and token-mixing components. Such a decomposition not only enhances computational efficiency but also provides a novel perspective in the context of approximation theory. Therefore, it is of theoretical interest to understand *how the combination of the token-wise and token-mixing operations contributes to the expressive power of models in handling sequential data.*

In this paper, we develop a general framework for the UAP analysis of transformers. Specifically, we provide abstract and verifiable conditions ensuring UAP, independent of specific architectural details. Our key contributions include:

- We derive a general sufficient condition for transformer models to achieve UAP in the $L^p$ sense over compact sets (Theorem 1), requiring: (1) feedforward layers satisfying the conditions [10], as stated in Definition 2, and (2) attention mechanisms producing distinct context-aware token representations across different inputs. Notably, our framework incorporates potential symmetry under token permutations in transformers, extending the analysis to non-transitive permutation groups beyond [45].

- For attention mechanisms that are analytic to their parameters, we reduce UAP verification to a two-sample condition (Theorem 2), simplifying practical validation compared to constructive approaches [37, 74–76]. Moreover, we show conditions under which transformers with a fixed number of attention layers but arbitrarily many feedforward layers achieve UAP, generalizing the results on the memorization capability of transformers in [37, 39].

- We apply our general framework to various transformer architectures to demonstrate its generality and applicability, including kernel-based [12, 65, 66], sparse [3, 40, 76], and some other attention mechanisms [9, 68]. For kernel-based attention (formulated in Section 4.1), our result (Corollary 1) provides UAP guarantee for many existing architectures in previous works [12, 65, 66] and also for new architectures. For sparse attention (e.g., architectures proposed in [3, 40, 76]), our result (Corollary 2) provides a UAP criterion which generalizes beyond the softmax attention and is free from technical assumptions on the sparse pattern.

- Our theoretical results also enable principled design of UAP-guaranteed architectures. We demonstrate this by proposing new transformer architectures with UAP guarantees, especially for attention mechanisms that preserve specific functional symmetry(Section 4.4).

We discuss in detail after our main results how they relate to the rest of the literature, and collect a more detailed review of related work in Section A.

---

[1]Here, we omit the layer normalization for simplicity of the analysis.

## 2 Problem formulation

In this section, we introduce the transformer-type architecture, an abstraction of the standard transformer [66] as a family of architectures composed of two repeating components: the token-mixing layers and token-wise map layers. Then, we define the universal approximation property (UAP). Notably, we introduce the UAP under permutation equivariance for any subgroup $G$ of the symmetric group $S_n$ over tokens, which is a more general framework.

In the following, we use $X = ([X]_1, \ldots, [X]_n) \in \mathbb{R}^{d \times n}$ to denote one data sample consisting of $n$ tokens $[X]_1, \cdots, [X]_n$ of dimension $d$. We say that $X$ is in *general position* if all of its tokens are distinct. We will also use the notation $[n] := \{1, \ldots, n\}$ for any positive integer $n$.

### 2.1 Transformer architecture

We present a general formulation for the two-step architecture of transformers as described in (1). The mapping $X_t \mapsto X_{t+1}$ can be abstracted as $(\mathrm{Id} + h) \circ (\mathrm{Id} + g)$. Here, $g$ generalizes the attention map $\mathrm{Atten}$ to a general token-mixing map, while $h$ generalizes the token-wise feedforward map FFN, which applies to $X \in \mathbb{R}^{d \times n}$ as:

$$h(X) := (\bar{h}([X]_1), \ldots, \bar{h}([X]_n)), \quad \text{where } \bar{h} : \mathbb{R}^d \to \mathbb{R}^d. \tag{2}$$

We denote by $\mathcal{G}$ a token-mixing family, consisting of functions mapping $\mathbb{R}^{d \times n}$ to $\mathbb{R}^{d \times n}$, to represent all possible choices of $g$ in a transformer. Recall that the attention layer in the original transformer [66] is given by

$$\mathrm{Atten}(X_t) = \sum_{i=1}^{N} W_V^{t,i} X_t \, \mathrm{softmax}\left( (W_K^{t,i} X_t)^\top W_Q^{t,i} X_t \right), \tag{3}$$

with trainable parameters $W_V^{t,i}, W_K^{t,i}, W_Q^{t,i} \in \mathbb{R}^{d \times d}$ for the $i$-th head in block $t$, and the softmax is applied column-wise. In this case, $\mathcal{G}$ is precisely the family of functions defined by (3) for all possible choices of $W_V^{t,i}, W_K^{t,i}, W_Q^{t,i}$.

Moreover, we consider

$$\mathcal{H}^{\otimes n} := \{ X \mapsto (\bar{h}([X]_1), \ldots, \bar{h}([X]_n)) \mid \bar{h} \in \mathcal{H} \}, \tag{4}$$

where $\mathcal{H}$ is a family of maps from $\mathbb{R}^d$ to $\mathbb{R}^d$, as the function family for the token-wise feedforward map $h$ in a transformer. We define a transformer block, the generalization of (1), to be a map in

$$\mathcal{F}_{\mathcal{G},\mathcal{H}} := \{ (\mathrm{Id} + h) \circ (\mathrm{Id} + g) \mid g \in \mathcal{G}, \ h \in \mathcal{H}^{\otimes n} \}. \tag{5}$$

A transformer identified by $\mathcal{G}$ and $\mathcal{H}$ is then the composition of such blocks, i.e. a map in the set:

$$\mathcal{T}_{\mathcal{G},\mathcal{H}} := \{ F_n \circ \cdots \circ F_1 \mid n \in \mathbb{N}, \ F_i \in \mathcal{F}_{\mathcal{G},\mathcal{H}} \}. \tag{6}$$

Notably, the feedforward layer can represent only tensor-type functions, i.e. functions of the form (2). The token-mixing mechanism extends this capability to more general functions by capturing the dependencies between tokens.

### 2.2 Universal approximation under permutation equivariance

Let $S_n$ denote the symmetric group on $n$ elements and let $G \leq S_n$ be a subgroup. Then, $G$ has a natural group action over $\mathbb{R}^{d \times n}$ by permuting the $d$-dimensional tokens. A function $f : \mathbb{R}^{d \times n} \to \mathbb{R}^{d \times n}$ is said to be $G$-*equivariant* if

$$f(\sigma(X)) = \sigma(f(X)), \quad \forall \sigma \in G, \ X \in \mathbb{R}^{d \times n}. \tag{7}$$

The original transformer and many of its variants have some degree of permutation equivariance over tokens. For instance, kernel-based token mixers [12, 66] typically have $G = S_n$, whereas sliding-window attention [3] employs a binary group (identity and reflection), and some architectures [68, 76] do not enforce any equivariance (i.e. $G = \{\mathrm{Id}\}$). If $\mathcal{G}$ consists of only $G$-equivariant functions, then $\mathcal{T}_{\mathcal{G},\mathcal{H}}$ can approximate only $G$-equivariant target functions. This motivates the following definition:

**Definition 1** ($G$-UAP). [2] *The transformer-type model with hypothesis space $\mathcal{T}_{\mathcal{G},\mathcal{H}}$ is said to have the $G$-universal approximation property (G-UAP) in the $L^p$ sense ($1 \leq p < \infty$) if, for every continuous $G$-equivariant function $F : \mathbb{R}^{d \times n} \to \mathbb{R}^{d \times n}$, every compact set $K \subset \mathbb{R}^{d \times n}$, and every $\varepsilon > 0$, there exists $\hat{F} \in \mathcal{T}_{\mathcal{G},\mathcal{H}}$ such that*

$$\|\hat{F} - F\|_{L^p(K)} < \varepsilon. \tag{8}$$

In applications, the equivariance restriction on the transformer is often addressed by introducing positional encoding [66] on tokens. From a theoretical perspective, previous works [34, 37, 74, 75] have shown that if a family $\mathcal{T}_{\mathcal{G},\mathcal{H}}$ has $G$-UAP for some $G$, then for any given compact set $K$, there exists an absolute positional encoding $\text{Enc} : X \to X + E$, where $E$ is a fixed matrix, such that

$$\mathcal{T}_{\mathcal{G},\mathcal{H}} \circ \text{Enc} := \{F \circ \text{Enc} \mid F \in \mathcal{F}_{\mathcal{G},\mathcal{H}}\} \tag{9}$$

can approximate any continuous function on $K$ in the $L^p$ sense without symmetry constraints. Technically, this can be done by making the domains of each token position distinct. On the other hand, there are also applications where exact symmetry needs to be enforced, such as structure-to-property prediction in crystals [8, 35, 60, 71]. Therefore, it is sufficient to consider the $G$-UAP, which can naturally extend to the general UAP while also covering cases where symmetry is considered. We will hereafter focus on the $G$-UAP.

In the literature, several works have studied the universal approximation of symmetric functions [13, 19, 45, 73], often focusing on specific architectures and symmetric groups. Notably, [45] provides a general sufficient condition for the action of any transitive subgroup of $S_n$ on coordinates (1-dimensional tokens). In comparison, this work considers approximation under symmetry in a general setting, with group action over $d$-dimensional tokens instead of coordinates. Additionally, our results apply to non-transitive group cases, which are not covered in [45]. In [1], the authors studied the ensemble controllability of control systems under symmetry, showing that systems that can interpolate arbitrarily many samples under symmetry are generic in a topology sense. However, this result does not tell us whether or not a given architecture has controllability. In comparison, our target is to provide a verifiable sufficient condition for UAP of specific architectures.

Our analysis focuses on fixed-length sequence-to-sequence maps on compact subsets. This setting directly covers encoder-style tasks and many architectural variants, which underlies many practical applications ranging from automatic speech recognition and visual sequence modeling to structure–property prediction in molecules and crystals [26, 35, 36, 58, 81]. In parallel, there are also measure-theoretic formulations in the literature that treat inputs as probability measures, which can handle variable or even infinite context length under continuity/regularity assumptions [20–22], offering complementary insights to our results.

## 3 Main results

In this section, we establish a general sufficient condition for the UAP of transformer-type architectures. Since transformer architectures consist of token-wise feedforward layers and token-mixing attention layers, we first provide conditions for each component required for UAP.

For the feedforward family $\mathcal{H}^{\otimes n}$, we introduce the following definition:

**Definition 2** (Nonlinearity and affine-invariance for $\mathcal{H}$). *We say a function family $\mathcal{H}$ (consisting of functions from $\mathbb{R}^d$ to $\mathbb{R}^d$) is nonlinear and affine-invariant, if*

- *For any $h \in \mathcal{H}$ and any $W, A \in \mathbb{R}^{d \times d}, b \in \mathbb{R}^d$, the function $Wh(A \cdot -b)$ also belongs to $\mathcal{H}$;*

- *$\mathcal{H}$ contains at least one non-affine Lipschitz function.*

The nonlinearity and affine-invariance condition holds for almost all practical feedforward layers, independent of specific choices of activation functions and the width of the network. When $d \geq 2$, according to the main result in [10], this condition ensures that the family

$$h \in (\text{Id} + \mathcal{H})^m = \{(\text{Id} + h_m) \circ \cdots \circ (\text{Id} + h_1) \mid h_1, \ldots, h_m \in \mathcal{H}\} \tag{10}$$

---

[2]Notice that in Definition 1, we do not require $\mathcal{T}_{\mathcal{G},\mathcal{H}}$ to consist of only $G$-equivariant maps, but only that it can approximate $G$-equivariant functions.

can approximate any continuous function $f : \mathbb{R}^d \to \mathbb{R}^d$ in $L^p$ sense over compact set. Therefore, this condition guarantees that only the token-wise feedforward layer is able to generate complex features over a single token.

However, an inherent limitation on the expressive power of feedforward layers is that they operate token-wise, meaning that they do not model any interactions between tokens. Considering this, we introduce the following definition for the attention family $\mathcal{G}$:

**Definition 3** (Token distinguishability for $\mathcal{G}$). *For a given group $G \leq S_n$ and a set of samples $D := \{X_i\}_{i=1}^N \subset \mathbb{R}^{d \times n}$ that are all in general position, we say a token-mixing family $\mathcal{G}$ can distinguish tokens in $D$ using $m$ layers under $G$-action, if there exists*

$$g \in (\mathrm{Id} + \mathcal{G})^m = \{(\mathrm{Id} + g_m) \circ \cdots \circ (\mathrm{Id} + g_1) \mid g_1, \ldots, g_m \in \mathcal{G}\} \tag{11}$$

*such that for any distinct $i, j \in [N]$ with $X_i$ and $X_j$ belonging to different orbits under the $G$-action (i.e., $X_i \neq \sigma(X_j)$ for all $\sigma \in G$), the tokens of $g(X_i)$ and $g(X_j)$ are all distinct.*

*Moreover, we say $\mathcal{G}$ satisfies the **token distinguishability condition** under $G$-action, if for any finite set $D$, there exists $m$ such that $\mathcal{G}$ can distinguish tokens in $D$ using $m$ layers under $G$-action.*

The token distinguishability condition ensures that token-mixing layers can model interactions between tokens by generating unique outputs for tokens in a finite set (up to $G$-action), enabling distinct in-context information for each token. This property is crucial for the expressive power of transformers, as illustrated below.

Consider a scenario where the token distinguishability condition fails: there exists a set $\Omega \in \mathbb{R}^{d \times n}$ with positive Lebesgue measure and some $i \in [n]$ such that $[g(X)]_i$ is constant over $\Omega$ for any $g \in (\mathrm{Id} + \mathcal{G})^m$. Consequently, any $F \in \mathcal{T}_{\mathcal{G}, \mathcal{H}}$ is also constant over $\Omega$, leading to the failure of UAP. This example shows that if too many tokens are indistinguishable by the token-mixing mechanism (e.g., from a positive measure set), the transformer's expressive power becomes limited.

On the other hand, the token distinguishability condition is relatively mild, as it only demands the composition of token-mixing layers to distinguish tokens, rather than enforcing any precise relation. This condition is generally easy to satisfy, provided $\mathcal{G}$ includes sufficiently diverse maps that can effectively mix tokens.

In the following, we assume that $d \geq 2$ and the zero map is in $\mathcal{G}$. Based on Definitions 2 and 3, we can state our first main result on the UAP of transformers:

**Theorem 1.** *Suppose that $\mathcal{H}$ is nonlinear and affine-invariant Definition 2, and $\mathcal{G}$ satisfies the token distinguishability condition Definition 3. Then, the family of transformers $\mathcal{T}_{\mathcal{G}, \mathcal{H}}$ satisfies the $G$-UAP Definition 1.*

Theorem 1 provides a general condition for the UAP of transformers. However, directly verifying the token distinguishability condition is challenging since we need to check the condition arbitrarily many times. Therefore, we propose the following theorem, which greatly simplifies the procedure.

**Theorem 2.** *We assume that $\mathcal{G}$ is parametrized by $\mathcal{G} = \{X \mapsto g(X; \theta) \mid \theta \in \Theta \subseteq \mathbb{R}^m\}$, where $\Theta$ is a connected open subset of $\mathbb{R}^m$, and for any fixed $X$, the mapping $\theta \mapsto g(X; \theta)$ is analytic. Then, if $\mathcal{G}$ can distinguish tokens of any dataset $D$ with two elements (Definition 3) using finite many layers, then $\mathcal{G}$ satisfies the token distinguishability condition.*

*Moreover, if there exists a uniform $m$ such that with $m$ layers, $\mathcal{G}$ can distinguish tokens of dataset $D$ with $|D| = 2$, then it can also do it for any finite dataset $D$ using $m$ layers. In this case, a deep model using only $m$ token-mixing layers and sufficiently many feedforward layers can achieve the UAP.*

The key insight in the proof of Theorem 2 is that if token distinguishability fails over a finite set, we can derive an equation in $\theta \in \Theta$ that is identically zero. By leveraging the property that the zero set of a non-trivial analytic function has measure zero, the equation can be reduced to the case of two elements, as detailed in Section B.2. The use of the analytic property is straightforward but significantly simplifies the token distinguishability condition.

Given expressive enough feedforward layers, Theorem 1 highlights the role of token-mixing mechanisms in transformer architectures for UAP: generating distinct, context-aware token representations. This aligns with prior works [37, 75, 76], which introduced "contextual mapping" to establish UAP for transformers. For instance, [37] defines "contextual mapping" as a function distinguishing tokens

in a dataset $D$ (similar to $g$ in Definition 3) without group actions. However, these works rely on explicit constructions, making verification complex and less generalizable. In contrast, Theorem 1 is the first to our knowledge that formulates token distinguishability and feedforward layer conditions as a general, non-constructive criterion for UAP. There is no need to explicitly construct for UAP once the conditions are verified. Additionally, Theorem 2 significantly simplifies the construction-based verification of token distinguishability, enabling broader applicability to diverse attention mechanisms, as shown in the examples in Section 4. Furthermore, the uniformity of $m$ in Theorem 2 also provides a convenient approach on the memorization capacity of attention layers studied in [37].

## 4 Applications to practical architectures

We demonstrate the generality and applicability of our UAP results by applying them to practical transformer architectures. We first follow the kernel-based framework from [65], which provides a unified description for a series of attention mechanisms. Specifically, many attention variants proposed in prior work can be formulated as

$$[\text{Atten}(X)]_i = \frac{\sum_{j \in \mathcal{N}(i)} k([W_Q X]_i, [W_K X]_j)[W_V X]_j}{\sum_{j \in \mathcal{N}(i)} k([W_Q X]_i, [W_K X]_j)}, \quad W_Q, W_K, W_V \in \mathbb{R}^{d \times d}, \quad (12)$$

where $k : \mathbb{R}^d \times \mathbb{R}^d \to \mathbb{R}^+$ is a positive kernel function, and $\mathcal{N}(i) \subset [n]$ denotes the set of indices that the $i$-th token attends to. In the original transformer, the kernel function is defined as $k(x, y) = \exp(x^\top y)$, and $\mathcal{N}(i) = [n]$.

Under this framework, many transformer variants can be categorized into two types, to which we will apply our results::

- **Kernel modification**: Replacing the kernel function $k$ to improve efficiency or performance. For example, using a kernel of the form $k(x, y) = \phi(x)^\top \phi(y)$ with a feature map $\phi : \mathbb{R}^d \to \mathbb{R}^m$ can significantly reduce computational cost when $m \ll n$.

- **Sparse attention**: For each $i$, restricting $\mathcal{N}(i)$ to a subset of $[n]$, reducing the number of tokens each token attends to. Here, we discuss in a general sense where $\mathcal{N}(i)$ can be dynamic across different layers, such as the sparse pattern in [40, 76].

### 4.1 Kernel-based attention

We first consider the kernel modification case, where we assume $\mathcal{N}(i) = [n]$ for all $i$. The following result follows from Theorem 1:

**Corollary 1.** *Suppose the kernel function $k$ satisfies the following conditions:*

- *$k(\cdot, \cdot) : \mathbb{R}^d \times \mathbb{R}^d \to \mathbb{R}^+$ is an analytic function.*

- *For any $x \in \mathbb{R}^d \setminus \{0\}$ and distinct points $y_1, y_2 \in \mathbb{R}^d \setminus \{0\}$, for almost all $W_K \in \mathbb{R}^{d \times d}$[3], the following holds:*

$$\lim_{t \to \infty} \frac{k(x, tW_K y_1)}{k(x, tW_K y_2)} = 0 \text{ or } +\infty. \quad (13)$$

*That is, for almost all given $W_K$, the kernel function $k$ can distinguish token representations by scaling the key vectors with a large factor.*

*Then, a transformer with kernel-based attention family $\mathcal{G}$ and feedforward family $\mathcal{H}$ satisfying the conditions in Theorem 1 possesses the $S_n$-UAP. Moreover, using only one token-mixing layer and sufficiently many feedforward layers can achieve the UAP.*

Corollary 1 ensures the distinguishability condition in Theorem 1 through the limiting behavior of the kernel function. This generalizes the idea from [74, 75], where softmax was used as an approximation of hardmax in explicit constructions. In comparison, our approach leverages analyticity, allowing the limit behavior to directly establish the distinguishability condition without further constructions.

Consequently, this result applies to various existing attention mechanisms. In particular, the following kernels directly satisfy the condition in Corollary 1:

---

[3]means that the condition holds all the whole space except for a measure zero set.

- $k(x, y) = \exp(x^\top y)$, used in the original transformer.
- $k(x, y) = \exp(-\gamma \|x - y\|_2^2)$ for $\gamma > 0$, the RBF kernel, explored in [65].
- $k(x, y) = \phi(x)^\top \phi(y)$, where

$$\phi(x)^\top = \exp\left(-\frac{1}{2}\|x\|^2\right) \left(\exp(\omega_1^\top x), \ldots, \exp(\omega_m^\top x)\right) \in \mathbb{R}^m, \tag{14}$$

with $\omega_1, \ldots, \omega_m \in \mathbb{R}^d$ being fixed weights drawn i.i.d. from a Gaussian distribution. This kernel is used in Performer [12], where Theorem 1 holds almost surely.

Among these, the UAP for the original transformer and Performer have already been shown [2, 74]. Our result recovers these results in our framework and relaxes the requirement on the architecture to achieve UAP: for original transformer, we do not need the bias in query vectors as in [74]; for Performer, we do not need additional hidden dimensions as in [2]. To the best of our knowledge, the UAP for RBF kernel attention is new, demonstrating the generality of our approach. Moreover, we can easily propose other kernels satisfying the condition in Theorem 1 but have not been studied in the literature, such as the following forms of $k(x, y)$:

- $k(x, y) = \exp(w^\top(x + y))$ for some $w \in \mathbb{R}^d \setminus \{0\}$;
- $k(x, y) = p(x - y)\tilde{k}(x, y)$, with $p$ being any positive polynomial function and $\tilde{k}$ being any kernel mentioned above.

Corollary 1 also generalizes the results in [37] on the memory capacity of transformers, where they prove that for transformers with dense softmax attention, one layer of attention is sufficient to achieve the UAP. Our result extends this to a broader class of kernel-based attention mechanisms.

## 4.2 Sparse attention

Prior works proposed sparse attention mechanisms to reduce the computational complexity of attention blocks [3, 11, 15, 24, 40, 76]. A common intuition for designing sparse patterns while retaining expressivity is ensuring connectivity, i.e., each token can attend to others via multiple "hops." For instance, in sliding window attention [3], where $\mathcal{N}(i) = \{j \in [n] \mid |j - i| \leq w\}$ with $w \ll n$, long-range interactions are achieved indirectly via multiple attention layers. In the following, we formalize this intuition and provide a general UAP condition for sparse attention transformers as a direct consequence of Theorem 1.

Denote $P([n])$ as the power set of $[n]$, i.e. the set of subsets of $[n]$. For a given function $\mathcal{N} : [n] \to P([n])$, we define $\mathcal{G}_\mathcal{N}$ as the family of maps from $\mathbb{R}^{d \times n} \to \mathbb{R}^{d \times n}$ defined by (12) associated with the sparsity pattern $\mathcal{N}$. We define the adjacency matrix of $\mathcal{N}$ as an $n \times n$ matrix $A_\mathcal{N}$ with $A_\mathcal{N}(i, j) = 1$ if $j \in \mathcal{N}(i)$ and $A_\mathcal{N}(i, j) = 0$ otherwise.

We also define

$$\mathrm{Aut}(\mathcal{N}) := \{\sigma \in S_N \mid j \in \mathcal{N}(i) \Leftrightarrow \sigma(j) \in \mathcal{N}(\sigma(i))\} \tag{15}$$

as the permutations that keep the structure of $\mathcal{N}$ invariant.

Let $\Phi := (\mathcal{N}_1, \mathcal{N}_2, \cdots, )$ be a sequence of sparsity patterns. We define the sparse transformer family associated with $\Phi$ as:

$$\mathcal{T}_\mathcal{H}^\Phi := \{(\mathrm{Id} + h_n) \circ (\mathrm{Id} + g_n) \circ \cdots \circ (\mathrm{Id} + h_1) \circ (\mathrm{Id} + g_1) \mid n \in \mathbb{N}_+, h_i \in \mathcal{H}^{\otimes n}, g_i \in \mathcal{G}_{\mathcal{N}_i}, \text{ for } i \in [n]\}. \tag{16}$$

Such a definition formulates transformers with dynamic sparse attention patterns.

**Definition 4.** *We call the sparsity pattern $\Phi$ to be connected within $m$ layers, if for any $i \neq j \in [n]$, there exists a sequence $1 \leq r_1 < r_2 < \cdots < r_k \leq m$ such that*

$$A_{\mathcal{N}_{r_k}} A_{\mathcal{N}_{r_{k-1}}} \cdots A_{\mathcal{N}_{r_1}}(i, j) > 0. \tag{17}$$

*That is, any token can reach any other token through a subsequence of the $m$ sparse attention layers.*

Also, let $\mathcal{H}$ be a family of token-wise feedforward layers satisfying the condition in Theorem 1, and $k$ be a kernel function satisfying the condition in Corollary 1. Then, we have the following result:

**Corollary 2.** *Suppose that $\Phi$ is connected within $m$ layers. Then, $\mathcal{T}_{\mathcal{F}}^{\Phi}$ possesses the G-UAP, where*

$$G = \bigcap_{i=1}^{\infty} \mathrm{Aut}(\mathcal{N}_i). \tag{18}$$

*Moreover, transformer with only $m$ layers of attention associated with the sparsity patterns $\mathcal{N}_1, \cdots, \mathcal{N}_m$ in $\Phi$ and sufficient number of token-wise feedforward layers can achieve the UAP.*

Corollary 2 provides a rigorous justification that the heuristic in keeping connectivity in the attention layers is also sufficient for UAP. Results from graph theory indicate that when $n$ is large, a random sparse pattern $\mathcal{N}$ has a trivial automorphism group with probability approaching 1 [17]. This fact indicates that with the guarantee of connectivity, most of the sparse attention patterns allow the UAP without symmetric restriction even in the absence of positional encodings.

Corollary 2 can cover many existing sparse attention mechanisms, including the following:

- the periodic pattern switching between "fixed attention" and "strided attention" in [11];
- the sliding window attention with/without global seeds (tokens connect to all others) in [3];
- the star-shape attention in [24], where one token attends to all others and the others connect in a circle;
- BigBird, a mixture of sliding windows, global seeds and random connections in [76].

The UAP of transformers with sparse softmax attention have also been studied in [75, 76] via a constructive approach. Compared to their results, Corollary 2 has several advantages. First, our results for UAP do not require other technical conditions, such as the periodicity of the sparse patterns and the existence of Hamiltonian path in [75], or each sparse pattern contains a star sub-structure in [76], other than the connectivity of the graph. For example, if the connection mode $\Phi$ is not periodic (e.g. there are different random patterns across layers), and some of the $\mathcal{N}_i$ do not contain a star graph mode, our result can still be applied as long as the connectivity is kept, while the results in [75, 76] may not be applicable.

In addition, our results can be applied to all kernels that satisfy the condition in Corollary 1, which generalizes the results based on explicit construction using softmax attention. Moreover, we identify the number of token-mixing layers required to achieve UAP as the minimal number of "hops" for each token to attend to all other tokens. In contrast, the results in [75, 76] use unbounded number of layers to achieve UAP. In this regard, our result can also be viewed as a generalization of the results on the minimal number of attention layers for UAP in [37] to sparse transformers.

### 4.3 Other attention mechanisms

Our framework can also be conveniently applied to many other variants of attention mechanisms that cannot be covered by (12). For example, the token distinguishability condition can be verified for the following architectures using similar method as in Corollary 1:

- Linformer [68], where the attention layer is defined as

$$\mathrm{Atten}(X) = W_V^t X F \, \mathrm{softmax}((W_K^t X E)^{\top} W_Q^t X), \tag{19}$$

  where $E, F \in \mathbb{R}^{n \times k}$ with $1 \leq k \ll n$ are two trainable projection matrices. This variant of attention reduces the complexity of attention from $\mathcal{O}(n^2)$ to $\mathcal{O}(nk)$.

- Kernelized attention used in SkyFormer [9], where the attention mechanism is given by:

$$[\mathrm{Atten}(X)]_i = \sum_{j=1}^{n} \exp\left(-\frac{1}{2}\|[W_Q X]_i - [W_K X]_j\|^2\right) W_V X_j. \tag{20}$$

**Corollary 3.** *We have that: (i) LinFormer satisfies the UAP without symmetric restriction; (ii) SkyFormer satisfies the $S_n$-UAP.*

The proofs are provided in Section C.4 using Theorem 2, similar to the proof of Corollary 1.

### 4.4 New attention mechanisms from the approximation analysis

Our results also provide insights into designing new transformer architectures with inherent UAP guarantees. In particular, our framework inspires the design of architectures with UAP under specific symmetries. In this section, we present examples of such designs to illustrate these insights.

#### 4.4.1 New attention mechanism with bias term

We propose a new architecture that naturally satisfies the conditions in Theorem 1 and Corollary 2. We consider the following attention mechanism with bias term:

$$[\text{Atten}(X)]_i = [X]_i + \sum_{j \in \mathcal{N}(i)} a\alpha(W[X]_j - b), \tag{21}$$

where $a \in \mathbb{R}, W \in \mathbb{R}^d$ and $b \in \mathbb{R}$ are learnable parameters. Assume that $\alpha$ is of polynomial growth, i.e. there exists $M$ and $N$ such that $|\alpha(x)| \le M(1 + |x|^N)$ for all $x \in \mathbb{R}$.

Then, the result in Corollary 2 still holds, if we replace the attention mechanism in (12) with (21). See Section C.4.3 for the formal statement and proof.

#### 4.4.2 Transformer with UAP under specific symmetry

In many applications, architectures with specific symmetric restrictions are required. Our framework also offers a new perspective on designing such architectures with UAP guarantees. For a given permutation group $G \le S_n$, we can design $G$-equivariant token-mixing layers that satisfy token distinguishability under $G$-action. By Theorem 1, a transformer with such token-mixing layers and a feedforward family $\mathcal{H}$ with nonlinearity and affine invariance achieves $G$-UAP. This simplifies the design process, as only token distinguishability is required for the token-mixing layer.

For some subgroups $G$ of $S_n$, the design for token-mixing layers can be very simple. Here, we use the example of $G = D_n$, the dihedral group of order $2n$, and the cyclic group $C_n$ of order $n$ to demonstrate. We identify $D_n \le S_n$ as the group generated by the cycle $\rho := (1, 2, \cdots, n)$ and the reflection $\sigma := (1, n)(2, n - 1) \cdots$. $C_n$ is the cyclic group generated by $\rho$. $D_n$ corresponds to the symmetry of a regular $n$-gon, relevant in applications like molecular structure [6, 23, 41]. Symmetry under $C_n$ applies to modeling periodic data, such as periodic time series [18, 27] and classifying periodic variable stars in cosmology [80].

For $D_n$, we provide the designs of token-mixing layers with token distinguishability.

**Architecture with $D_n$-symmetry**  Choose the token-mixing layers defined in (12) or in (21) with the sparsity pattern

$$\mathcal{N}(i) := \{(i + j) \bmod n \mid j = -w, \cdots, 0, \cdots, w\}, \tag{22}$$

where $w \le \lfloor \frac{n-1}{2} \rfloor - 1$ is an integer. Here, we assume that the kernel $k$ in (12) and the function $\alpha$ in (21) satisfy the conditions in Corollaries 1 and 5, respectively.

For $C_n$, similar designs as above also work. However, we can use a simpler one based on convolution:

**Architecture with $C_n$-symmetry:**  Define the token-mixing layer via column-wise convolution:

$$\text{Atten}(X) = \psi * X, \quad \text{where } [\psi * X]_i = \sum_{j=0}^{l} \psi_j [X]_{(\ell+j) \bmod n}, \quad i = 0, 1, \ldots, n - 1. \tag{23}$$

with a trainable kernel $\psi = [\psi_0, \cdots, \psi_l] \in \mathbb{R}^{l+1}$ for some integer $l \ge 1$.

This design can be viewed as an adaptation of the temporal convolutional network [33, 43], treating the input sequence as a circular structure. The following statement holds, with proof in Section C.4.3:

**Corollary 4.** *The architecture with $D_n$-symmetry and $C_n$-symmetry defined above satisfies the token distinguishability condition under the action of $D_n$ and $C_n$, respectively. Moreover, the transformer with such token-mixing mechanisms and a non-linear affine-invariant family $\mathcal{H}$ possesses the $D_n$-UAP or $C_n$-UAP, respectively.*

Architectures incorporating symmetry through convolutional layers have been studied in the literature [13, 14, 46, 70, 80]. In particular, [80] also proposed a convolutional structure for representing $C_n$-invariant functions, although it does not inherently guarantee UAP. In contrast, our proposed architecture naturally satisfies the $C_n$-UAP property according to Theorem 1. Moreover, by choosing specific sparse mode $\mathcal{N}$, we can generalize the architecture for $D_n$ symmetry to other permutation subgroups of $S_n$ that can be identified as the automorphism group of an order-$n$ directed graph. For more general permutation groups, our framework can still be applied if one can find token-mixing layer satisfying the token distinguishability condition under the action of the group. The method proposed in [31, 61] may be helpful in identifying such layers. We believe our framework provides a new perspective on the design of equivariant/invariant architectures with UAP guarantees.

## 5 Conclusion

In this paper, we investigate the universal approximation property (UAP) of general transformer architectures within a unified framework. Our main results, Theorem 1 and Theorem 2, provide general and verifiable conditions for establishing UAP across a range of attention-based architectures, avoiding complex constructions as in previous works. This generality is demonstrated in Section 4, where we apply our framework to various attention types. Moreover, our results offer guidance for designing new attention mechanisms with UAP guarantees, as illustrated in Section 4.4. We also acknowledge certain limitations of this work. First, normalization layers commonly used in practice are not considered, and extending our analysis to incorporate them would be valuable. Second, some architectures, such as those in [38], do not satisfy the analyticity assumption in Theorem 2. Although the condition in Theorem 1 remains verifiable for such architectures, it remains unclear whether our results on the required number of token-mixing layers for UAP still hold. Moreover, as our results offer non-constructive yet verifiable criteria for UAP—abstracting away the specific forms of token-mixing and token-wise modules—they do not yield quantitative insight into the relative contributions of each architectural component. A systematic, quantitative characterization of how individual mechanisms (e.g., multi-head attention, mixture-of-experts, low-rank projections) affect approximation efficiency remains an important direction for future work.

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

## A  Detailed related works

**Approximation results for transformers**   Since the introduction of the transformer architecture in [66], numerous studies have investigated its approximation properties. The universal approximation property (UAP) of the original transformer with softmax attention as a fixed length sequence-to-sequence model was established in [74]. This constructive approach was later extended to transformers with certain sparse attention mechanisms [75, 76]. In [37], the authors proposed a new construction demonstrating that transformers with a single attention layer and sufficiently deep feedforward networks can achieve UAP. Similarly,[2] showed that with increased hidden dimensions, two variants of transformers, i.e. LinFormer [68] and Performer [12], satisfy the UAP. In contrast to these constructive methods for different architectures, our results offer a unified framework for establishing the UAP of various transformer models without relying on explicit constructions. Beyond these works treating transformers as a fixed length sequence-to-sequence model, there are also studies handling transformers with variable-length inputs by considering the input sequence as an empirical measure [20–22]. The universal interpolation and universal approximation properties under this viewpoint were established with proper assumptions. Compared to these results, we still consider transformers as sequence-to-sequence models in this work to offer a direct analysis for different transformer architectures. Recently, there are other works studying the UAP of attention-only architectures [29, 48], indicating that softmax attention alone can also achive strong approximation power. Other studies have explored the UAP of transformers under alternative settings, such as in-context learning, prompting, and constrained scenarios [20, 30, 42, 50, 56]. Another researchline is the Turing completeness of transformers [55, 69]. Besides these UAP results, there are also works providing the approximation rates of transformers. For instance,[34] provides explicit rates over a dense subset of sequence-to-sequence functions; [67] derives rates for target functions with structured memory; and [64] characterizes the approximation rate in terms of function smoothness for transformers with infinitely long inputs.

**Approximation under symmetry**   The study of approximation under functional symmetries has been explored in various works. In [73], the universal approximation of functions invariant under compact group or translation actions was analyzed using shallow neural networks. In [13, 19], convolutional structures were proposed to approximate equivariant functions. In [45, 52, 59], the universal approximation under symmetry using deep neural networks was investigated. Notably, [45] provides a general sufficient condition for the action of any transitive subgroup of $S_n$ on coordinates (1-dimensional tokens). In contrast, our work addresses approximation under symmetry in a broader setting, considering group actions on $d$-dimensional tokens rather than coordinates, thereby extending the analysis in [45] to non-transitive permutation groups. In [1], the authors provide a general framework on the ensemble controllability of control systems under symmetry. They also show that systems that can interpolate arbitrarily many samples under symmetry are generic in certain topology. Compare to their genericity results, our results provide a verifiable sufficient condition for UAP of specific architectures, allowing direct applications to various transformer architectures.

**Transformer variants**   Beyond the original Transformer, numerous architectural variants have been developed to improve efficiency, scalability, or adaptability. These include sparse attention mechanisms [3, 15, 24, 40, 76], low-rank and kernel-based approximations of attention [12, 68, 72], as well as other architectural modifications [25, 47, 51, 77]. Another related line of work explores parameter-efficient fine-tuning methods for large Transformer models [28, 32, 49, 53, 78, 79], which aim to adapt pretrained networks to downstream tasks with minimal additional parameters. In this paper, we establish a general sufficient condition for the universal approximation property (UAP) of various Transformer variants. As demonstrated in Section 4, our framework can be readily verified for many existing architectures, and potentially extended to other designs not included as well.

## B  Proof of Theorem 1 and Theorem 2

### B.1  Proof of Theorem 1

In the following, $\| \cdot \|_2$ denotes the $\ell^2$-norm, for both vectors in $\mathbb{R}^{d \times n}$ or $\mathbb{R}^d$. We begin by proving the following interpolation property of $\mathcal{T}_{\mathcal{G}, \mathcal{H}}$:

**Proposition 1** (**Interpolation Property**). *Suppose $\mathcal{G}$ and $\mathcal{H}$ satisfy the condition in Theorem 1 for group G. Consider any G-equivariant continuous function $F : \mathbb{R}^{d \times n} \to \mathbb{R}^{d \times n}$. Then, for any $\varepsilon > 0$ and $\{X_i\}_{i=1}^N \subset \mathbb{R}^{d \times n}$, there exists $\hat{F} \in \mathcal{T}_{\mathcal{G},\mathcal{H}}$ such that:*

- *$\|\hat{F}(X_i) - F(X_i)\|_2 < \varepsilon$, if $X_i$ is in general position(defined in Section 2).*

- *$\|\hat{F}(X_i)\|_2 < n \cdot \max_i\{\|F(X_i)\|_2\} + 2\varepsilon$, if $X_i$ is not in general position.*

*Proof.* Since $F$ and functions in $\mathcal{T}_{\mathcal{G},\mathcal{H}}$ are G-equivariant, we only need to consider the case when $X_i$ are from distinct orbits under the G-action. Moreover, we can assume that $X_i$ are in general position for $i = 1, \cdots, M$, and $X_i$ are not in general position for $i = M+1, \cdots, N$.

By the token distinguishability condition, there exist $m$ and $g \in (\mathrm{Id} + \mathcal{G})^m$ such that the tokens of $g(X_i)$ are all distinct for $i = 1, \cdots, M$. We denote $x_1, \cdots, x_{Mn}$ as the distinct tokens in $g(X_1), \cdots, g(X_M)$. For each $j \leq Mn$, suppose $x_j = [X_l]_k$ for some $(l, k) \in [n] \times [M]$, we denote $y_j$ as its corresponding token $[F(X_l)]_k$. Moreover, we denote $x_{Mn+1}, \cdots, x_{Mn+J} \in \mathbb{R}^d$ as the distinct tokens in $g(X_{M+1}), \cdots, g(X_N)$ that are different from $x_1, \cdots, x_{Mn}$. For each $j > Mn$, we denote $y_j = 0 \in \mathbb{R}^d$. Then, we get a set of $d$-dimensional pairs $\{(x_j, y_j)\}_{j=1}^{Mn+J} \subset \mathbb{R}^d \times \mathbb{R}^d$ where all $x_j$ are all distinct. Since $\mathcal{H}$ satisfies the non-linear affine invariance condition, according to the main result in [10], we know that there exists a function

$$f \in \{(\mathrm{Id} + f_k) \circ \cdots \circ (\mathrm{Id} + f_1) \mid k \in \mathbb{N}_+, f_1, \cdots, f_k \in H\} \tag{24}$$

such that

$$\|f(x_j) - y_j\|_2 \leq \frac{1}{n}\varepsilon, \quad j = 1, \cdots, Mn + J. \tag{25}$$

Denote $f^{\otimes n}$ as the token-wise extension of $f$ to $\mathbb{R}^{d \times n}$. For each $i \leq M$, we have

$$\|f^{\otimes n}(g(X_i)) - F(X_i)\|_2 = \left\| \left( \big(f(g[X_i]_1) - [F(X_i)]_1\big), \cdots, \big(f(g[X_i]_n) - [F(X_i)]_n\big) \right) \right\|_2 \tag{26}$$
$$\leq n \cdot \max_j \|f(x_j) - y_j\|_2 \leq \varepsilon.$$

For each $i \geq M + 1$, we have

$$\|f^{\otimes n}(g(X_i))\| = \left\| \left( f(g[X_i]_1), \cdots, f(g[X_i]_n) \right) \right\|_2$$
$$\leq \sum_{l:g[X_i]_l \in \{x_1, \cdots, x_{Mn}\}} \|f(g[X_i]_l)\|_2 + \sum_{l:g[X_i]_l \in \{x_{Mn+1}, \cdots, x_{Mn+J}\}} \|f(g[X_i]_l)\|_2$$
$$\leq n \cdot \max_j \|f(x_j)\|_2 + n \cdot \max_{j>Mn} \|f(x_j)\|_2$$
$$\leq n \cdot \left( \max_i\{\|F(X_i)\|_2\} + \frac{\varepsilon}{n} \right) + \varepsilon = n \cdot \max_i\{\|F(X_i)\|_2\} + 2\varepsilon. \tag{27}$$

Therefore, $\hat{F} = f^{\otimes n} \circ g \in \mathcal{T}_{\mathcal{G},\mathcal{H}}$ satisfies the interpolation property. $\qquad\square$

**Proposition 2** (**Approximation of Tensor-Type Functions**). *Suppose $\mathcal{G}$ and $\mathcal{H}$ satisfy the condition in Theorem 1 for group G. Consider any continuous, increasing function $h : \mathbb{R} \to \mathbb{R}$ and any $\varepsilon > 0$. Then, there exists $F \in \mathcal{T}_{\mathcal{G},\mathcal{H}}$ such that*

$$\|F - h^{(d \times n)}\|_{C(K)} \leq \varepsilon, \tag{28}$$

*where $h^{(d \times n)}$ is the coordinate-wise extension of $h$ to $\mathbb{R}^{d \times n}$, given by*

$$(h^{(d \times n)}(X))_{ij} = h(X_{ij}), \quad 1 \leq i \leq d,\ 1 \leq j \leq n. \tag{29}$$

*Proof.* Proposition 2 directly follows from the proof of Proposition 4.11 [44] and Theorem 2.6 in [10]. $\qquad\square$

**Proposition 3** (Corollary of Main results of [10]). *Let $d \geq 2$ and $\mathcal{H}$ be a family of maps from $\mathbb{R}^d$ to $\mathbb{R}^d$ that satisfies the non-linearity and affine-invariance condition in Theorem 1. Then, for any $\{(x_i, y_i)\}_{i=1}^N \subset \mathbb{R}^d \times \mathbb{R}^d$ with $x_i \neq x_j$ for all $i \neq j$ and $\varepsilon > 0$, there exists*

$$f \in \{(\mathrm{Id} + f_k) \circ \cdots \circ (\mathrm{Id} + f_1) \mid k \in \mathbb{N}_+, f_1, \cdots, f_k \in \mathcal{H}\} \tag{30}$$

*such that*

$$\|f(x_i) - y_i\| \leq \varepsilon, \quad i = 1, \cdots, N. \tag{31}$$

*Proof.* Proposition 3 is a direct corollary of Theorem 2.6 in [10]. $\qquad\square$

*Proof of Theorem 1.* The approach of the proof is similar to the main theorem in [44] and [45].

We assume without loss of generality that $K = [-s, s]^{d \times n}$ is a hypercube in $\mathbb{R}^{d \times n}$. Our target is to show that for any $\varepsilon > 0$, we can find function $\hat{F} \in \mathcal{T}_{\mathcal{G}, \mathcal{H}}$ such that $\|\hat{f} - F\|_{L^p(K)} \leq \varepsilon$.

**Step 1.** For each multi-index $\mathbf{i} = (i_{kl})_{k \in [d], l \in [n]} \in \mathbb{Z}^{d \times n}$ and $\delta > 0$, we define the grid cells:

$$\square_{\mathbf{i}, \delta} := \left\{ X \in \mathbb{R}^{d \times n} \mid X_{kl} \in [i_{kl}\delta, (i_{kl} + 1)\delta], \text{ for all } k \in [d], l \in [n] \right\}. \tag{32}$$

We denote $\boldsymbol{p}_{\mathbf{i}, \delta} := \delta\mathbf{i}$ as a corner point of $\square_{\mathbf{i}, \delta}$, and $\chi_{\mathbf{i}, \delta}$ as the characteristic function of $\square_{\mathbf{i}, \delta}$. Since $F$ is continuous, there exists $\delta > 0$ such that it has a piece-wise constant approximation

$$\tilde{F} := \sum_{\mathbf{i}} F(\boldsymbol{p}_{\mathbf{i}}) \chi_{\mathbf{i}, \delta}, \tag{33}$$

such that

$$\|F - \tilde{F}\|_{L^p(K)} \leq \frac{\varepsilon}{4}. \tag{34}$$

**Step 2.** Apply Proposition 3 to the set of grid points

$$\left\{ \boldsymbol{p}_{\mathbf{i}, \delta} \mid \mathbf{i}_{kl} \in \left\{ \lceil -s/\delta \rceil, \ldots, \lfloor s/\delta \rfloor \right\} \right\}.$$

Then, for any $\gamma > 0$, there exists a function $\bar{F} \in \mathcal{T}_{\mathcal{G}, \mathcal{H}}$ such that:

- If $\boldsymbol{p}_{\mathbf{i}, \delta}$ is in general position, then

$$\|\bar{F}(\boldsymbol{p}_{\mathbf{i}, \delta}) - F(\boldsymbol{p}_{\mathbf{i}, \delta})\| < \gamma.$$

- Otherwise,

$$\|\bar{F}(\boldsymbol{p}_{\mathbf{i}, \delta})\| < n \cdot \max_{i}\{\|F(\boldsymbol{p}_{\mathbf{i}, \delta})\|\} + 2\gamma \leq n\|F\|_{C(K)} + 2\gamma.$$

**Step 3.** For any $\alpha \in (0, 1)$, define

$$\square_{\mathbf{i}, \delta}^{\alpha} := \left\{ X \in \mathbb{R}^{d \times n} \mid X_{kl} \in [i_{kl}\delta, i_{kl}\delta + \alpha\delta], \text{ for all } k \in [d], l \in [n] \right\}, \tag{35}$$

as the shrunk hypercube of $\square_{\mathbf{i}, \delta}$ with side length $\alpha\delta$. We consider the map $h_{\alpha, \delta} : \mathbb{R} \to \mathbb{R}$ defined as:

$$h_{\alpha, \delta}(x) = \begin{cases} i\delta, & \text{if } x \in [i\delta, i\delta + \alpha\delta], \quad i \in \mathbb{Z}, \\ i\delta + \dfrac{x - i\delta - \alpha\delta}{1 - \alpha}, & \text{if } x \in [i\delta + \alpha\delta, (i+1)\delta], \quad i \in \mathbb{Z}. \end{cases} \tag{36}$$

On the interval $[i\delta, i\delta + \alpha\delta]$, the function $h_{\alpha, \delta}(x)$ remains constant at $i\delta$. Then, as $x$ increases from $i\delta + \alpha\delta$ to $(i+1)\delta$, the function increases linearly from $i\delta$ to $(i+1)\delta$. Notice that $h_{\alpha, \delta}^{(d \times n)}$ is continuous and has constant value $\boldsymbol{p}_{\mathbf{i}}$ on $\square_{\mathbf{i}, \delta}^{\alpha}$ for each $\mathbf{i}$. Since $h_{\alpha, \delta}$ is continuous and increasing, by Proposition 2, for any $\rho > 0$, there exists a function $H_{\alpha, \delta} \in \mathcal{T}_{\mathcal{G}, \mathcal{H}}$ such that

$$\|H_{\alpha, \delta} - h_{\alpha, \delta}^{(d \times n)}\|_{C(K)} < \rho. \tag{37}$$

**Step 4.** Now we can estimate the error of the composition $\|\bar{F} \circ H_{\alpha, \delta} - F\|_{L^p(K)}$. We define

$$K^{\alpha} = \left( \bigcup_{\mathbf{i}} \square_{\mathbf{i}, \delta}^{\alpha} \right) \cap K,$$

and denote $K_1^{\alpha}$ as the union of the grid cells in $K^{\alpha}$ whose corner point $\boldsymbol{p}_{\mathbf{i}}$ are in general position, and $K_2^{\alpha}$ as the union of the grid cells in $K^{\alpha}$ whose $\boldsymbol{p}_{\mathbf{i}}$ are not in general position. Then, we have

1. On $K_1^\alpha$, we have

$$\|\bar{F} \circ h_{\alpha,\delta}^{(d\times n)} - \tilde{F}\|_{L^p(K_1^\alpha)} < \gamma(m(K_1^\alpha))^{\frac{1}{p}}. \tag{38}$$

2. For $K_2^\alpha$, the number of $\mathbf{i}$ with $\boldsymbol{p}_{\mathbf{i},\delta}$ not in general position is at most $\frac{n(n-1)}{2}(2s/\delta+1)^{(n-1)d}$, we have the measure of $K_2^\alpha = \mathcal{O}(\delta^d)$. Therefore,

$$\|\bar{F} \circ h_{\alpha,\delta}^{(d\times n)} - \tilde{F}\|_{L^p(K_2^\alpha)} < (n \cdot \|F\|_{C(K)} + 2\gamma)\mathcal{O}(\delta^d). \tag{39}$$

Therefore, we can choose $\delta$ and $\gamma$ sufficiently small such that

$$\|\bar{F} \circ h_{\alpha,\delta}^{(d\times n)} - \tilde{F}\|_{L^p(K^\alpha)} < \frac{\varepsilon}{4}. \tag{40}$$

On $K \setminus K^\alpha$, by choosing $\alpha$ sufficiently close to 1, we make $m(K \setminus K^\alpha)$ arbitrarily small. Since $\bar{F} \circ h_{\alpha,\delta}^{(d\times n)}$ and $\tilde{F}$ is bounded on $K$, the following can be guaranteed:

$$\|\bar{F} \circ h_{\alpha,\delta}^{(d\times n)} - \tilde{F}\|_{L^p(K\setminus K^\alpha)} < \frac{\varepsilon}{4}. \tag{41}$$

Since $\bar{F}$ is uniformly continuous on $K$, there exists $\rho > 0$ such that for any $X, Y \in K$ with $\|X - Y\| < \rho$, we have $\|\bar{F}(X) - \bar{F}(Y)\| < \kappa := \varepsilon/(4(m(K))^{\frac{1}{p}})$. Therefore, after determining $\alpha, \delta$ and $\gamma$, we can choose $\rho$ such that

$$\|\bar{F} \circ H_{\alpha,\delta} - F\|_{L^p(K)} \le (m(K))^{\frac{1}{p}}\kappa \le \frac{\varepsilon}{4}. \tag{42}$$

After determining $\delta$ and $\alpha$, by step 3, we can choose $\rho$ sufficiently small such that

$$\|\bar{F} \circ H_{\alpha,\delta} - \bar{F} \circ h_{\alpha,\delta}^{(d\times n)}\|_{L^p(K)} \le \rho \operatorname{Lip}(\bar{F}) < \frac{\varepsilon}{4}. \tag{43}$$

Recall also that

$$\|\tilde{F} - F\|_{L^p(K)} < \frac{\varepsilon}{4}.$$

Thus, by the triangle inequality,

$$\begin{aligned} \|\bar{F} \circ H_{\alpha,\delta} - F\|_{L^p(K)} &\le \|\bar{F} \circ H_{\alpha,\delta} - \bar{F} \circ h_{\alpha,\delta}^{(d\times n)}\|_{L^p(K)} + \|\bar{F} \circ h_{\alpha,\delta}^{(d\times n)} - \tilde{F}\|_{L^p(K^\alpha)} \\ &\quad + \|\bar{F} \circ h_{\alpha,\delta}^{(d\times n)} - \tilde{F}\|_{L^p(K\setminus K^\alpha)} + \|\tilde{F} - F\|_{L^p(K)} \\ &< \frac{\varepsilon}{4} + \frac{\varepsilon}{4} + \frac{\varepsilon}{4} + \frac{\varepsilon}{4} = \varepsilon, \end{aligned} \tag{44}$$

which completes the proof. $\qquad\square$

### B.2  Proof of Theorem 2

*Proof.* Assume that Condition 2 in Theorem 1 fails. Then, there exists $N$ samples $\{X_i\}_{i=1}^N$ from different orbits under the $G$-action, but for any $m$ and $g \in (\operatorname{Id} + \mathcal{G})^m$, there exist indices $i, j \in [N]$ such that at least one token in $g(X_i)$ is identical to a token in $g(X_j)$. This can be written as

$$\Pi_{i,j} := \prod_{l_1,l_2} \|[g(X_i)]_{l_1} - [g(X_j)]_{l_2}\|_2^2 = 0. \tag{45}$$

When $\mathcal{G}$ is parametric in $\theta \in \Theta$, $\Pi_{i,j}$ is also analytic in $\theta$. As the zero set of a nonzero real analytic function has measure zero [54], this implies that for some $i, j$, $\Pi_{i,j}$ is identically zero, meaning that $\mathcal{G}$ fails to distinguish tokens in $X_i$ and $X_j$. This argument reduces Condition 2 in Theorem 1 to the case $N = 2$.

Moreover, if there exists a uniform $m$ such that the token distinguishability condition for any two tokens holds, the above argument essentially shows that this $m$ can also be used to distinguish any $N$ tokens. $\qquad\square$

# C  Proofs for the applications of main results

## C.1  Proof of Corollary 1 and Corollary 2

We first prove the following lemma.

**Lemma 1.** *Let $r$ be a kernel function satisfying the conditions in Corollary 1. Suppose $\{a_1, \ldots, a_r\} \subset \mathbb{R}^d \setminus \{0\}$ and $\{b_1, \ldots, b_s\} \subset \mathbb{R}^d \setminus \{0\}$ are two sequences of distinct tokens. Suppose that for some indices $r' \in [r]$ and $s' \in [s]$, the following equality holds for all choices of $W_Q, W_K, W_V$:*

$$a_{r'} + \sum_{j=1}^{r} \left( \frac{k(W_Q a_{r'}, W_K a_j)}{\sum_{l=1}^{r} k(W_Q a_{r'}, W_K a_l)} W_V a_j \right) = b_{s'} + \sum_{j=1}^{s} \left( \frac{k(W_Q b_{s'}, W_K b_j)}{\sum_{l=1}^{s} k(W_Q b_{s'}, W_K b_l)} W_V b_j \right). \quad (46)$$

*Then, $r = s$, and there exists a permutation $\sigma \in S_n$ with $\sigma(r') = s'$ such that $a_i = b_{\sigma(i)}$ for all $i$.*

*Proof.* Taking $W_V$ to be zero directly gives $a_{r'} = b_{s'}$. In the following, we set $W_Q = I$, and for notational simplicity define

$$\tilde{k}(x) := k(a_{r'}, x), \quad \forall x \in \mathbb{R}^d.$$

The proof utilizes the following lemma:

**Lemma 2** (Auxiliary lemma). *Let $\{x_1, \cdots, x_p\} := \{a_1, \cdots, a_r\} \cup \{b_1, \cdots, b_s\}$. Then, there exist $W_K \in \mathbb{R}^{d \times d}$ and a permutation $\sigma \in S_p$ such that*

$$\lim_{t \to \infty} \frac{\tilde{k}(tW_K x_{\sigma(j)})}{\tilde{k}(tW_K x_{\sigma(i)})} = \infty, \text{ for all } i < j. \quad (47)$$

Since $\{a_j\}_{j=1}^{r}$ are $r$ distinct tokens, by the auxiliary lemma, we may choose a $W_K$ (and reindex the sequences accordingly) so that after replacing $W_K$ by $tW_K$ the kernel values satisfy, for large $t > 0$,

$$\tilde{k}(tW_K a_1) \ll \tilde{k}(tW_K a_2) \ll \cdots \ll \tilde{k}(tW_K a_r),$$

and similarly

$$\tilde{k}(tW_K b_1) \ll \tilde{k}(tW_K b_2) \ll \cdots \ll \tilde{k}(tW_K b_s).$$

Moreover, the lemma tells us that for any $a_j$ and $b_l$, either $a_j = b_l$ or one of $\tilde{k}(tW_k a_j)$ and $\tilde{k}(tW_k b_l)$ is dominated by the other in the limit $t \to \infty$.

After subtracting $a_{r'}$ from both sides in (46), we have

$$\frac{\sum_{j=1}^{r} \tilde{k}(tW_K a_j) a_j}{\sum_{j=1}^{r} \tilde{k}(tW_K a_j)} = \frac{\sum_{j=1}^{s} \tilde{k}(tW_K b_j) b_j}{\sum_{j=1}^{s} \tilde{k}(tW_K b_j)}. \quad (48)$$

By a transformation, it gives

$$\sum_{j=1}^{r} \sum_{l=1}^{s} \tilde{k}(tW_K a_j) \tilde{k}(tW_K b_l)(a_j - b_l) = 0. \quad (49)$$

Let $t \to \infty$, considering the dominate term $\tilde{k}(tW_K a_j)\tilde{k}(tW_K b_l)(a_r - b_s)$ gives $a_r = b_s$. Then, for all $q = 0, 1, \cdots, \min\{r, s\}$, we then prove by induction that: $a_{r-k} = b_{s-k}$.

Suppose we already have $a_{r-i} = b_{s-i}$, for $i = 1, \cdots, q-1$. It then follows that

$$\sum_{j=r-q+1}^{r} \sum_{l=s-q+1}^{s} \tilde{k}(tW_K a_j)\tilde{k}(tW_K b_l)(a_j - b_l)$$

$$= \sum_{j=r-q+1}^{r} \sum_{l=r-q+1}^{r} \tilde{k}(tW_K a_j)\tilde{k}(tW_K a_l)(a_j - a_l) = 0, \quad \text{for all } t \in \mathbb{R}. \quad (50)$$

.

Combining with (49), we have

$$(\sum_{j=1}^{r-q}\sum_{l=1}^{r-q}+\sum_{j=1}^{r-q}\sum_{l=s-q+1}^{s}+\sum_{j=r-q+1}^{r}\sum_{l=1}^{s-q})\left(\tilde{k}(tW_K a_j)\tilde{k}(tW_K b_l)(a_j - b_l)\right) = 0, \qquad (51)$$

where the leading term is

$$\tilde{k}(tW_K a_{r-q})\tilde{k}(tW_K b_s)(a_{r-q} - b_s) + \tilde{k}(tW_K a_r)\tilde{k}(tW_K b_{s-q})(a_r - b_{s-q}). \qquad (52)$$

Since $a_r = b_s$, $a_{r-q} \neq a_r$ and $b_{r-q} \neq b_r$, let $t \to \infty$ gives that $\tilde{k}(tW_K a_{r-q})$ and $\tilde{k}(tW_K b_{s-q})$ are not dominated by each other. By the auxiliary lemma, this indicates that $a_{r-q} = b_{s-q}$, which completes the induction.

Then, we have shown that $a_{r-k} = b_{s-k}$ for all $k = 0, 1, \cdots, \min\{r, s\}$. The only remaining thing is to show that $r = s$. Suppose $r < s$, then we have

$$\sum_{j=1}^{r}\sum_{l=1}^{s-r}\tilde{k}(tW_K a_j)\tilde{k}(tW_K b_l)(a_j - b_l) \equiv 0, \qquad (53)$$

where the unique leading term is $\tilde{k}(tW_K a_r)\tilde{k}(tW_K b_{s-r})(a_r - b_{s-r})$. Since $a_r = b_r \neq b_{s-r}$, that gives a contradiction. This completes the proof. $\qquad\square$

*Proof of the auxiliary lemma.* For each pair $(i, j)$ with $1 \leq i < j \leq p$, we define

$$\mathcal{M}_{i,j} := \mathbb{R}^{d \times d} \setminus \left\{ W_K \,\middle|\, \lim_{t \to \infty} \frac{\tilde{k}(tW_K x_i)}{\tilde{k}(tW_K x_j)} = 0 \text{ or } \infty \right\}, \qquad (54)$$

and we define $\mathcal{M}$ as the union of all such sets:

$$\mathcal{M} = \bigcup_{1 \leq i < j \leq p} \mathcal{M}_{i,j}. \qquad (55)$$

According to the condition in Corollary 1, each $\mathcal{M}_{i,j}$ is a measure-zero set in $\mathbb{R}^{d \times d}$. Therefore, $\mathcal{M}$ is also measure-zero. Choose any $W_K \in \mathbb{R}^{d \times d} \setminus \mathcal{M}$. Then, for any $i, j$, we have

$$\lim_{t \to \infty} \frac{\tilde{k}(tW_K x_i)}{\tilde{k}(tW_K x_j)} = 0 \text{ or } \infty, \qquad (56)$$

i.e. either $\tilde{k}(tW_K x_i) \ll \tilde{k}(tW_K x_j)$ or $\tilde{k}(tW_K x_i) \gg \tilde{k}(tW_K x_j)$ for large $t$. This indicates that there exists a permutation $\sigma \in S_p$ such that

$$\lim_{t \to \infty} \frac{\tilde{k}(tW_K x_{\sigma_1(j)})}{\tilde{k}(tW_K x_{\sigma_1(i)})} = \infty, \text{ for all } i < j, \qquad (57)$$

which completes the proof. $\qquad\square$

***Proof of Corollary 1.*** Lemma 1 shows the token-distinguishability condition over $X$ with non-zero tokens. This indicates the interpolation property over the region when $X$ has non-zero tokens. Since the set

$$\{X \in \mathbb{R}^{d \times n} \mid [X]_i = 0 \text{ for some } i \in [n]\} \qquad (58)$$

is a measure-zero set, the $S_n$-UAP holds for $\mathcal{T}_{\mathcal{G}, \mathcal{H}}$ by the same argument as Theorem 1. $\qquad\square$

## C.2 Verification of conditions in Corollary 1 on practical attention mechanisms

We verify the conditions in Corollary 1 for the following kernel functions:

- $k(x, y) = \exp(x^\top y)$, used in the original transformer [66].
  **Verification**: For any given $x \neq 0$ and distinct $y_1, y_2 \in \mathbb{R}^d \setminus \{0\}$, the set

$$\mathcal{P} := \{W_K \mid x^\top W_K y_1 = x^\top W_K y_2\} \qquad (59)$$

is a hyperplane in $\mathbb{R}^{d \times d}$, which has zero measure. Notice that for any $W_K$ in $\mathbb{R}^{d \times n} \setminus \mathcal{P}$,

$$\lim_{t \to \infty} \frac{\exp(tx^\top W_K y_1)}{\exp(tx^\top W_K y_2)} = \lim_{t \to \infty} \exp(t(x^\top W_K(y_1 - y_2))) = \infty \text{ or } 0. \tag{60}$$

This indicates that the condition in Corollary 1 holds.

- $k(x, y) = \exp(-\gamma \|x - y\|_2^2)$, the RBF kernel, explored in [65].
  **Verification:** Notice that
  $$\frac{\exp(-\gamma \|x - tW_K y_1\|_2^2)}{\exp(-\gamma \|x - tW_K y_2\|_2^2)} = \exp(-\gamma(\|tW_K y_1\|_2^2 - \|tW_K y_2\|_2^2) + 2tx^\top W_K(y_2 - y_1)). \tag{61}$$

  Therefore, any $W_K$ such that $\|W_K y_1\|_2 \neq \|W_K y_2\|_2$ satisfies the condition in Corollary 1. Since for distinct $y_1$ and $y_2$,
  $$\|W_K y_1\|_2^2 - \|W_K y_2\|_2^2 = 0, \tag{62}$$
  is a non-zero quadratic equation on $W_K$, whose solution set has zero measure. Therefore, the condition in Corollary 1 holds.

- $k(x, y) = \phi(x)^\top \phi(y)$, where
  $$\phi(x)^\top = \exp\left(-\frac{1}{2}\|x\|^2\right) \left(\exp(\omega_1^\top x), \ldots, \exp(\omega_m^\top x)\right) \in \mathbb{R}^m, \tag{63}$$

  with $\omega_1, \ldots, \omega_m \in \mathbb{R}^d$ drawn i.i.d. from a Gaussian distribution. This kernel is used in Performer [12], and Corollary 1 holds almost surely in this case.
  **Verification:** We have
  $$\begin{aligned}
  \frac{k(x, tW_K y_1)}{k(x, tW_K y_2)} &= \frac{\phi(x)^\top \phi(tW_K y_1)}{\phi(x)^\top \phi(tW_K y_2)} \\
  &= \exp\left(\frac{t^2}{2}(\|W_K y_2\|_2^2 - \|W_K y_1\|_2^2)\right) \frac{\sum_{i=1}^m \exp(\omega_i^\top x)\exp(t\omega_i^\top W_k y_1)}{\sum_{i=1}^m \exp(\omega_i^\top x)\exp(t\omega_i^\top W_k y_2)}
  \end{aligned} \tag{64}$$

  We claim that if $w_i$ are pair-wise linear independent, i.e. there does not exist $i \neq j$ such that $w_i = \alpha w_j$ for some $\alpha \in \mathbb{R}$, the condition in Corollary 1 holds. This almost surely holds when $w_i$ are drawn i.i.d. from a Gaussian distribution.
  If when $t \to \infty$, the ratio in (64) do not goes to infinity or zero, it must hold that
  $$\|W_K y_1\|_2 = \|W_K y_2\|_2, \text{ and } \max_i\{\omega_i^\top W_K y_1\} = \max_i\{\omega_i^\top W_K y_2\}. \tag{65}$$

  When $w_i$ are pair-wise linear independent, we have that $y_1 w_i^\top \neq y_2 w_j^\top$ for all $i \neq j \in [m]$. Also, since $y_1 \neq y_2$ and $w_i$ are non-zero(by the pair-wise independent condition), we have $y_1 w_i^\top \neq y_2 w_i^\top$ for all $i \in [m]$. Therefore, we have that all the sets
  $$\{W_K \mid \omega_i^\top W_K y_1 = \omega_j^\top W_K y_1\} = \{W_K \mid \langle W_K, y_1 w_i^\top - y_2 w_j^\top \rangle_F = 0\}, \tag{66}$$

  where $\langle \cdot, \cdot \rangle_F$ denotes the Frobenius inner product, are hyperplanes in $\mathbb{R}^{d \times d}$, which has zero measure. That is, equation (65) only holds for a measure-zero set of $W_K$, which completes the verification.

- $k(x, y) = \exp(w^\top x) + \exp(w^\top y)$, where $w \in \mathbb{R}^d$.
  **Verification:** We have
  $$\frac{k(x, tW_K y_1)}{k(x, tW_K y_2)} = \frac{\exp(w^\top(x + tW_K y_1))}{\exp(w^\top(x + tW_K y_2))} = \exp(tw^\top W_K(y_1 - y_2)). \tag{67}$$

  When $y_1 \neq y_2$, for almost all $W_K$, we have $w^\top W_K(y_1 - y_2) \neq 0$. Therefore, the condition in Corollary 1 holds.

- $k(x, y) = p(x - y)\tilde{k}(x, y)$, with $p$ being any positive polynomial function and $\tilde{k}$ being any kernel satisfies the condition in Corollary 1.
  **Verification:** Just neet to notice that for almost all $W_k$, it holds that
  $$\lim_{t \to \infty} \frac{p(x - tW_K y_1)}{p(x - ptW_K y_2)} \tag{68}$$
  is a constant indicating that the condition in Corollary 1 still holds.

## C.3 Proof of Corollary 2

*Proof.* We only need to prove the token distinguishability condition for two samples:

- For any $X$ and $Y$ that are in general positions and from different orbits of $G$(defined in (18)), there exists

$$g \in \mathcal{G}_m^\Phi := \{(\mathrm{Id} + g_m) \circ \cdots \circ (\mathrm{Id} + g_1) \mid g_i \in \mathcal{G}_{\mathcal{N}_i}, \text{ for } i \in [m]\} \tag{69}$$

such that the tokens of $g(X)$ and $g(Y)$ are all distinct.

We prove this claim by contradiction. Assume that, there exist $X$ and $Y$ that are in general positions and from different orbits of $G$, but for any $g \in \mathcal{G}_m^\Phi$, there exist indices $i, j \in [n]$ such that at least one token in $g(X)$ is identical to a token in $g(Y)$. Then, according to the analyticity, there exist indices $i_1$ and $i_2$ such that, $[g(X)]_{i_1} = [g(Y)]_{i_2}$ always hold. For a given $p_1 \in [m]$, we first consider $g \in \mathrm{Id} + \mathcal{G}_{\mathcal{N}_{p_1}}$. Then, $[g(X)]_{i_1} = [g(Y)]_{i_2}$ gives:

$$[X]_{i_1} + \frac{\sum_{j \in \mathcal{N}_{p_1}(i_1)} k([W_Q X]_{i_1}, [W_K X]_j)[W_V X]_j}{\sum_{j \in \mathcal{N}_{p_1}(i_1)} k([W_Q X]_{i_1}, [W_K X]_j)} = [Y]_{i_2} + \frac{\sum_{l \in \mathcal{N}_{p_1}(i_2)} k([W_Q Y]_{i_2}, [W_K Y]_l)[W_V Y]_l}{\sum_{j \in \mathcal{N}_{p_1}(i_2)} k([W_Q Y]_{i_2}, [W_K Y]_l)},$$
$$\tag{70}$$

for any $W_Q, W_K, W_V \in \mathbb{R}^{d \times d}$. According to Lemma 1, we can deduce that $|\mathcal{N}_{p_1}(i)| = |\mathcal{N}_{p_1}(j)|$, $[X]_{i_1} = [Y]_{i_2}$, and

$$\{[X]_q \mid q \in \mathcal{N}_{p_1}(i_1)\} = \{[Y]_q \mid q \in \mathcal{N}_{p_1}(i_2)\}. \tag{71}$$

Therefore, choose any $q_1 \in \mathcal{N}_{p_1}(i_1)$, we can find $q_2 \in \mathcal{N}_{p_1}(i_2)$ such that $[X]_{q_1} = [Y]_{q_2}$.

Now, we claim that for any $p_2 < p_1$ and $g \in \mathrm{Id} + \mathcal{G}_{\mathcal{N}_{p_2}}$ (a layer before the $p_1$-th layer) , $[g(X)]_{q_1} = [g(Y)]_{q_2}$. Otherwise, suppose there exists $g_1 \in \mathrm{Id} + \mathcal{G}_{p_2}$ with $[g_1(X)]_{q_1} \neq [g_1(Y)]_{q_2}$. By scaling the $W_V$ matrix to be small enough, we can assume $g_1$ satisfies that

$$\|g_1(X) - X\|_2 < \frac{1}{2} \min_{i \neq j}\{\|[X]_i - [X]_j\|_2\}. \tag{72}$$

Then, we have that $[g_1(X)]_{q_1} \neq [g_1(Y)]_{q_2}$, and by equation (72), for any $q \neq q_2$ in $\mathcal{N}(i_2)$, we have

$$\|[g_1(X)]_{q_1} - [g_1(Y)]_q\|_2 = \|([g_1(X)]_{q_1} - [X]_{q_1}) + ([X]_{pq1} - [Y]_q) + ([Y]_q - [g_1(Y)]_q)\|_2$$
$$\geq \|[X]_{g_1} - [Y]_q\|_2 - \|[g_1(X)]_{q_1} - [X]_{q_1}\|_2 - \|[Y]_q - [g_1(Y)]_q\|_2 > 0. \tag{73}$$

Therefore, $[g_1(X)]_{q_1}$ does not appear in the tokens of $g_1(Y)$, i.e. the sets

$$\{[g_1(X)]_j \mid j \in \mathcal{N}(i_1)\} \quad \text{and} \quad \{[g_1(Y)]_l \mid l \in \mathcal{N}(i_2)\} \tag{74}$$

must be different. By applying Lemma 1 to $[g_1(X)]_{q_1}$ and $[g_1(Y)]_{q_2}$, we know that there exists $g_2 \in \mathrm{Id} + \mathcal{G}_{\mathcal{N}_{p_1}}$ such that $[g_2(g_1(X))]_{q_1} \neq [g_2(g_1(Y))]_{q_2}$. Since $g_2 \circ g_1 \in (\mathrm{Id} + \mathcal{G}_{\mathcal{N}_{p_2}}) \circ (\mathrm{Id} + \mathcal{G}_{\mathcal{N}_{p_1}}) \subset \mathcal{G}_m^\Phi$ which can distinguish $[X]_{i_1}$ and $[Y]_{i_2}$, contradicting to our assumption.

Hence, we have shown that for $p_2 < p_1 \leq n$ and any $g \in \mathrm{Id} + \mathcal{G}_{\mathcal{N}_{p_2}}$, $[g(X)]_{q_1} = [g(Y)]_{q_2}$. Then, we can apply Lemma 1 to $[X]_{q_1}$ and $[Y]_{q_2}$, and deduce that the set of tokens of $X$ with indices in $\mathcal{N}_{p_s}(q_1)$ are the same as those of $Y$ with indices in $\mathcal{N}_{p_3}(q_2)$. That is, the tokens where $[X]_{i_1}$ can attend to within two hops are the same as those where $[Y]_{i_2}$ can attend to within two hops.

The above process can be repeated. Since we assume that $\Phi$ is connected within $m$ layers, we know that any indices in $[n]$ can be reached starting from $i_1$ and $i_2$ within $m$ hops. Finally, the above discussion can cover all indices in $[n]$. Since $X$ and $Y$ are in general positions, the correspondence between their tokens is unique. Finally, this results in a permutation $\sigma \in S_n$, such that:

$$[X]_i = [Y]_{\sigma(i)}. \tag{75}$$

On the other hand, apply Lemma 1 again to eqch $\mathcal{N}_p$, $[X]_i$ and $[Y]_{\sigma(i)}$, we can deduce that

$$j \in \mathcal{N}_p(i) \Leftrightarrow \sigma(j) \in \mathcal{N}_p(\sigma(i)), \text{ for all } p \in [m]. \tag{76}$$

By the definition of $\mathrm{Aut}(\mathcal{N})$, this means that $\sigma$ belongs to each $\mathcal{N}_p$, indicating that $\sigma \in G$. However, this contradicts to our assumption that $X$ and $Y$ are from different orbits of $\mathrm{Aut}(\Gamma)$, which completes the proof. $\square$

## C.4 Verification of the UAP for other transformer variants

In this section, we verify the condition in Theorem 1 for the kernelized attention of SkyFormer [9] and the attention mechanism of the Linformer [68].

### C.4.1 UAP for LinFormer

Linformer [68], where the attention layer is defined as

$$\text{Atten}(X) = X + W_V X F \, \text{softmax}((W_K X E)^\top W_Q X) \tag{77}$$

where $E, F \in \mathbb{R}^{n \times k}$ with $1 \le k \ll n$ are two trainable projection matrices.

For LinFormer, we have the following lemma:

**Lemma 3.** *Let $X, Y \in \mathbb{R}^{d \times n}$ be two points that are in general positions. If for some $i_1, i_2 \in [n]$, the following equality holds for all $W_Q, W_K, W_V \in \mathbb{R}^{d \times d}$ and $E, F \in \mathbb{R}^{n \times k}$:*

$$[X]_{i_1} + [W_V X F \, \text{softmax}((W_K X E)^\top W_Q X)]_{i_1} = [Y]_{i_2} + [W_V Y F \, \text{softmax}((W_K Y E)^\top W_Q Y)]_{i_2}, \tag{78}$$

*then we have $i_1 = i_2$ and $X = Y$.*

*Proof.* Take $W_V = 0$ gives $[X]_{i_1} = [Y]_{i_2}$. Then, for any given $R = \{r_1, \cdots, r_k\} \subset [n]$, we take

$$E = F = [e_{r_1}, \cdots, e_{r_k}], \tag{79}$$

where $e_{r_i}$ is the $r_i$-th column of the identity matrix. Equation (78) then gives that

$$\sum_{j \in R} \left( \frac{\exp(\langle W_Q[X]_{i_1}, W_K[X]_j \rangle)}{\sum_{l \in R} \exp(\langle W_Q[X]_{i_1}, W_K[X]_l \rangle)} W_V[X]_j \right) = \sum_{j \in R} \left( \frac{\exp(\langle W_Q[Y]_{i_2}, W_K[Y]_j \rangle)}{\sum_{l \in R} \exp(\langle W_Q[Y]_{i_2}, W_K[Y]_l \rangle)} W_V[Y]_j \right) \tag{80}$$

which reduces to the discussion in Lemma 1. Therefore, we have that the set $\{[X]_i \mid i \in R\}$ is the same as $\{[Y]_i \mid i \in R\}$. Since $X$ and $Y$ are in general positions, and $R$ is arbitrary, this indicates that $X = Y$. $\qquad\square$

According to this lemma and the fact that (19) is analytic to all the parameters, we conclude by Theorem 2 that the UAP holds for LinFormer without symmetric restrictions. Furthermore, the same result can be generalized to the case where the softmax function in (19) is replaced by a kernel-based form with a kernel satisfying the condition in Corollary 1.

### C.4.2 UAP for SkyFormer

The kernelized attention used in SkyFormer [9], where the attention mechanism is given by:

$$[\text{Atten}(X)]_i = [X]_i + \sum_{j=1}^n \exp\left( -\frac{1}{2} \|[W_Q X]_i - [W_k X]_j\|^2 \right) W_V X_j. \tag{81}$$

The proof follows from the verification for the RBF kernel Section C.2, with the same argument to prove the token-distinguishability condition.

### C.4.3 UAP for architecture proposed in (21)

More precisely, if we define $\tilde{\mathcal{G}}_\mathcal{N}$ as the family of token-mixing maps associated with the sparsity pattern $\mathcal{N}$, and the transformer family $\tilde{\mathcal{T}}_\mathcal{H}^\Phi$ associated with a sequence of sparse mode, just as $\mathcal{T}_\mathcal{H}^\Phi$ defined in Section 4.2. Then, under the same assumption on $\Phi$ as in Corollary 2, we have:

**Corollary 5.**

$\tilde{\mathcal{T}}_\mathcal{H}^\Phi$ possesses the $G$-UAP with $G$ defined in (18).

Assume the condition in Theorem 2 fails. Then, there exists $X$ and $Y$ that are in general positions, and $i_1, i_2 \in [n]$ such that for all $W \in \mathbb{R}^{d \times d}$, $a \in \mathbb{R}$ and $b \in \mathbb{R}^d$, we have

$$[X]_{i_1} + \sum_{j \in \mathcal{N}(i_1)} a\alpha(W[X]_j - b) = [Y]_{i_2} + \sum_{j \in \mathcal{N}(i_2)} a\alpha(W[Y]_j - b). \tag{82}$$

It then follows that $[X]_{i_1} = [Y]_{i_2}$. Moreover, if the sets

$$\{[X]_j \mid j \in \mathcal{N}(i_1)\} \tag{83}$$

and

$$\{[Y]_j \mid j \in \mathcal{N}(i_2)\} \tag{84}$$

are not the same, we can then derive an identity

$$\sum_{l=1}^{L} c_l \alpha(W x_l - b) = 0, \tag{85}$$

where $x_l$ are the unique tokens appears in the two sets, $c_i$ equals to 1 if $x_l$ appears in the set of $X$, and $-1$ otherwise. Notice that the identity holds for all $W \in \mathbb{R}^{d \times d}$ and $b \in \mathbb{R}$, this gives a contradiction according to the proof of Theorem 1 in [10]. Specifically, since $\alpha$ is of polynomial growth, taking the Fourier transform(in distributional sense) on both side of (85) with respect to $b$ gives:

$$(\sum_{l=1}^{L} c_l e^{i(W x_l)^\top \xi}) \hat{\alpha}(\xi) = 0, \quad \text{for all } \xi \in \mathbb{R}^d, \tag{86}$$

where $\hat{\alpha}$ is the Fourier transform of $\alpha$. Since $\alpha$ is not a polynomial, we have $\operatorname{supp} \hat{\alpha}$ contains a non-zero value. This will lead to a contradiction. See Section 3.2 in [10] for more details.

### C.4.4 Details for architecture proposed for $D_n/C_n$ equivariant map

**For architecture with $D_n$ symmetry** By choosing $\Phi = (\mathcal{N}, \mathcal{N}, \cdots)$ as an invariant sequence of sparse mode, by Corollary 2, we have that the transformer with the first design satisfies the $\operatorname{Aut}(\mathcal{N})$-UAP. Therefore, we only need to prove thet $\operatorname{Aut}(\mathcal{N}) = D_n$.

First, it is easy to see that for any $g \in D_n$ , we have $j \in \mathcal{N}(i)$ indicates that $g(j) \in \mathcal{N}(g(i))$. This implies that $N_n \in \operatorname{Aut}(\mathcal{N})$. On the other hand, for any $g \in \operatorname{Aut}(\mathcal{N})$, since $D_n$ is transitive, there exists $h \in D_n$ such that $h(g(1)) = 1$. Now, we prove that $\gamma := h \circ g$ is either the identity or the reflection $\sigma = (1, n)(2, n-1) \cdots$.

Since $\gamma \in \operatorname{Aut}(\mathcal{N})$, we know that $\gamma(\mathcal{N}(i)) = \mathcal{N}(\gamma(i))$. Since $\gamma(1) = 1$, we have $\gamma(\mathcal{N}(1)) = \mathcal{N}(\gamma(1)) = \mathcal{N}(1)$ is invariant. This indicates that for any $i \in \mathcal{N}(1)$, $\gamma(i)$ is also in $\mathcal{N}(1)$. Now, we consider the value of $\mathcal{N}(2)$. We have that

$$2w = |\mathcal{N}(1) \cap \mathcal{N}(2)| = |\gamma(\mathcal{N}(1)) \cap \gamma(\mathcal{N}(2))| = |\mathcal{N}(1) \cap \mathcal{N}(\gamma(2))| \tag{87}$$

Since $2w + 1 \leq n - 2$, we have that in $\mathcal{N}(1)$, there are only two indices $j = n, 2$ such that

$$|\mathcal{N}(1) \cap \mathcal{N}(j)| = 2w. \tag{88}$$

Therefore, it follows that $\gamma(2) = n$ or 2. If $\gamma(2) = 2$, we can then repeat the discussion to deduce that $\gamma(i) = i$ for $i = 2, 3 \cdots, n$ sequentially, indicating that $\gamma$ is the identity. If $\gamma(2) = n$. Then, we can repeat the discussion to deduce that $\gamma(i) = n + 2 - i \mod n$ for $i = 2, 3 \cdots, n$. This indicates that $\gamma$ is a reflection $(2, n)(3, n-1) \cdots$, which is in $D_n$.

Therefore, we have shown that $\operatorname{Aut}(\mathcal{N}) = D_n$. This is actually a classical result on the automorphism group of the circulant graph [4].

Moreover, if we destroy the symmetry to reflection by defining

$$\mathcal{N}(i) := \{i, i+1, i+2, \cdots, i+w \mod n\} \text{ for } i = 1, 2, \cdots, n, \tag{89}$$

with $w \leq \lfloor \frac{n-1}{2} \rfloor - 1$, we get a transformer that is $C_n$-equivariant and satisfies the $C_n$-UAP. For the proof, we only need to check that $\operatorname{Aut}(\mathcal{N}) = C_n$, which can be done following the same approach as $D_n$.

**For architecture with $C_n$ symmetry** For token-mixing layer defined by the convolution in (23), we first notice that it satisfies the $C_n$-equivariance. In fact, this follows from the fact that the convolutional operation is equivariant to translation.

Therefore, to prove the $C_n$-UAP, we only need to check the token distinguishability condition under $C_n$ action. Specifically, suppose the condition in Theorem 2 fails. Then, there exists $X$ and $Y$ that

are in general positions, and $i_1, i_2 \in [n]$ such that the composition of token mixing layers cannot distinguish the $i_1$-th token of $X$ and the $i_2$-th token of $Y$.

Considering using single layers, we have that for all $\psi \in \mathbb{R}^{l+1}$, it holds

$$[\psi * X]_{i_1} = [\psi * X]_{i_2}, \tag{90}$$

i.e.

$$\sum_{j=0}^{l} \psi_j \, [X]_{(i_1+j) \bmod n} = \sum_{j=0}^{l} \psi_j \, [Y]_{(i_2+j) \bmod n}, \tag{91}$$

which indicates that

$$\sum_{j=0}^{l} \psi_j \left( [X]_{(i_1+j) \bmod n} - [Y]_{(i_2+j) \bmod n} \right) = 0 \tag{92}$$

Since $\psi$ is arbitrary, this indicates that

$$[X]_{(i_1+j) \bmod n} = [Y]_{(i_2+j) \bmod n}, \quad \text{for } j = 0, 1, \cdots, l. \tag{93}$$

Then, we takethe indicices $i_1 + l \bmod n$ and $i_2 + l \bmod n$ of $X, Y$ respectively, and consider using two layers. The process is essentially the same as the proof of Corollary 2. We can finally deduce that

$$[X]_{(i_1+j) \bmod n} = [Y]_{(i_2+j) \bmod n}, \quad \text{for } j = 0, 1, \cdots, n. \tag{94}$$

That is, $X$ and $Y$ differs only a cyclic action on tokens, meaning that they are from the same $C_n$ orbit, which is a contradiction, and completes the proof.

