# OpenReview forum: "A unified framework for establishing the universal approximation of transformer-type architectures"
_NeurIPS.cc/2025/Conference — NeurIPS 2025 poster_

### Official Review · Reviewer_gCVj · 2025-07-02

**Clarity:** 3
**Significance:** 3
**Originality:** 2
**Rating:** 5
**Confidence:** 5

**Summary:**

The authors study the universal approximation property of transformer models (a variety of variants, including some not necessarily addressed in previous works), and prove a non-constructive result using the analyticity of the attention kernel (due to the softmax).

**Questions:**

1. Results that are independent of context length should also be referenced. This connects to the measure-theoretic perspective on transformers; see the works of Furuya, de Hoop and Peyré, as well as Geshkovski, Rigollet and Ruiz-Balet. In my opinion, this is also the more natural point of view. We are at a stage where the actual question in approximation theory is not entirely clear: yes, it is possible to obtain rates and approximate functions, but why is this meaningful? Moreover, the results are not constructive, which in my view misses the point. Indeed, one of the goals of the constructive methods would be to understand the role of different mechanisms (multi head, mixture of experts, etc.) to the inner workings of Transformers. I did not see this being elucidated in the present paper. If the authors comment on how the different mechanisms affect approximation proofs, I believe the quality of the paper would increase significantly.

2. I do not really understand why the authors work with functions on compact sets, as the topic is perhaps fundamentally about probability measures (given the permutation equivariance due to attention and the presence of positional encoding) and maps between spaces of probability measures. This is particularly true due to the fact that the context length $n$ is not necessarily fixed in practical applications.

3. Can the authors comment if there is a way to obtain approximation results with a complexity that is independent of the number of data points?

**Ethical Concerns:**

["NO or VERY MINOR ethics concerns only"]

**Final Justification:**

After the detailed answers by the authors, and consulting some of the other comments they made to the other referees, I am now convinced that this paper merits acceptance.

**Limitations:**

yes

**Quality:**

3

**Strengths And Weaknesses:**

The issue I have with this paper is that the formalism is not especially elegant mathematically, and the results are not particularly innovative. The distinguishability aspect has already been addressed in an abstract way by Agrachev and Letrouit in 2024, where they use the notions of strata and orbifolds, which are well known in the context of quotient manifolds. Perhaps I am mistaken, but this should be discussed even if I am.

I would be very open to changing my score, but the presentation of the paper needs to change, to something more fundamental, since this is a very theoretical paper anyway.

---

> ### Author Rebuttal · Authors · 2025-07-29
>
> We sincerely thank the reviewer for the thoughtful critique.  Below we address each concern in turn and will integrate the requested citations and clarifications in the future version.
>
> ## Relation to Agrachev and Letrouit (2024)
>
> We appreciate the reviewer pointing us to this important reference and will include it in the revised version. While both works study expressivity via symmetry-aware controllability, there are substantial differences:
>
> * **Notion of “distinguishability”**: In Agrachev & Letrouit, “distinct samples” refer to a set of distinct points $X_1,\cdots, X_N$ in the quotient manifold $M = \mathbb{R}^{d \times n} / G$. This is part of the problem formulation for the simultaneous controllability. In contrast, our notion of *token distinguishability* (Def. 3) is a condition imposed on the *architecture family* $\mathcal G$, requiring the existence of $g\in \mathcal G$ such that all the $d$-dimensional tokens in $g(X_1),\cdots, g(X_N)$ are distinct. This requirement arises from the token-wise implementation of the feedforward layers.
>
> * **Genericity vs. sufficient conditions**: Agrachev & Letrouit (2024) proves that the architectures that satisfy controllability is large in a topological sense - roughly that if we were to randomly (assuming that this could be done) pick a architecture, it should satisfy controllability. However, such results cannot tell us whether a specific architecture has controllability or not. This is where our type of results come in, providing explicit and verifiable sufficient conditions on transformer architectures to achieve approximation.
>
>     Both their and our results reveal the power of deep network approximation, but in different ways and complement our understanding.
>
>
> * **Approximation vs. interpolation**: Their notion of ensemble controllability aligns with interpolation property (Prop. 1 in Appendix A). In contrast, our results also incorporates the approximation results. As shown in Cheng, Li, Lin & Shen, 2025, interpolation does not imply UAP in general.
>
> ## On the measure-theoretic perspective and variable-length inputs
>
> We acknowledge the growing interest in framing transformers as maps between probability measures (e.g., Furuya et al., Geshkovski et al.), especially for variable-length inputs. Our work, however, focuses on *fixed-length* sequence-to-sequence transformers, which remain prevalent in practice (e.g., in encoding tasks, classification, and contrastive learning). In such settings, function approximation over compact subsets of $\mathbb{R}^{d \times n}$ remains standard.
>
> Furthermore, sparse, low-rank and some other attention mechanisms do not lift transparently to measure-valued inputs as the standard attention layer in Furuya et al; our framework accommodates them directly. By contrast, measure-theoretic perspectives should be important in formulating varying-length autoregressive decoders but may be less suited to these architectural variants.
>
> ## On the role of architectural mechanisms
> Our framework abstracts the transformer block into *token-mixing* and *token-wise* components, allowing us to analyze a wide class of architectures without relying on the exact form of architecture components, including multi-head attention, mixture-of-experts, etc. These mechanisms can be viewed as design choices within either component. While quantifying their precise roles in approximation is indeed an important direction, such analysis is orthogonal to our current goal of providing general UAP conditions and is left to future work.
>
> ## On approximation complexity
>
> We are not entirely certain what is meant by “complexity independent of the number of data points.” If the reviewer is referring to the sequence length $n$ (as commonly done in measure-theoretic works), then we believe such independence is unlikely. As the number of tokens increases, the space of functions to approximate becomes essentially more complex. For example, in interpolation-based frameworks like Agrachev & Letrouit (2024), the number of required number of samples $N$ grows exponentially with $n$ to support an approximation result.
>
> On the other hand, if the question concerns whether the approximation complexity can be independent of the number of interpolation points in the function domain, then Cheng, Li, Lin & Shen (2025) discusses related results in the feedforward case. Combining such insights with structure-specific analysis of token-mixing layers may lead to bounds in certain settings, but this is beyond the scope of our current work.
>
> We hope these clarifications address the reviewer’s concerns. We will incorporate the suggested references and improve the exposition, particularly with regard to clarity and positioning relative to prior work. Thank you again for your willingness to reconsider your evaluation.
>
> References:
>
> Geshkovski, B., Rigollet, P., & Ruiz-Balet, D. (2024). Measure-to-measure interpolation using Transformers. arXiv preprint arXiv:2411.04551.
>
> Furuya, T., de Hoop, M. V., & Peyré, G. Transformers are Universal In-context Learners. In The Thirteenth International Conference on Learning Representations.
>
> Cheng, J., Li, Q., Lin, T., & Shen, Z. (2025). Interpolation, approximation, and controllability of deep neural networks. SIAM Journal on Control and Optimization, 63(1), 625-649.
>
> Agrachev, A., & Sarychev, A. (2020). Control in the spaces of ensembles of points. SIAM Journal on Control and Optimization, 58(3), 1579-1596.

---

> > ### Comment · Reviewer_gCVj · 2025-08-05
> >
> > I thank the authors for their detailed reply.
> >
> > What I meant by "complexity independent of the number of data points" is the following: there are N sequences, each composed of n tokens. The question is: can the complexity of doing interpolation (which is often necessary for doing approximation) be less than linear in N. Clearly, for both approximation and interpolation, the number of tokens n can be taken to infinity (as demonstrated in the work of Furuya, de Hoop and Peyré). This should be clearly indicated in the paper.
> >
> > I also do not agree with the authors with regard to the comment "Furthermore, sparse, low-rank and some other attention mechanisms do not lift transparently to measure-valued inputs as the standard attention layer in Furuya et al; our framework accommodates them directly. By contrast, measure-theoretic perspectives should be important in formulating varying-length autoregressive decoders but may be less suited to these architectural variants."
> > In fact, I believe the opposite is the case: the measure-theoretical perspective is more valuable in encoder-only architectures due to the permutation equivariance, and less so in autoregressive models (where a conditional probability and Markov chain interpretation is more valuable).
> > Furthermore, all questions of sparsity and low-rankness can be easily modeled in the measure-theoretic case (by making assumpitons on the support of the measure..)
> > The measure-theoretic aspect is not here to undervalue the relevance of fixed context lengths. But rather to probe the true fundamentals and limitations of Transformers (e.g., what is the true relevance of finite context length, beyond obvously, applications). I believe that such questions warrant discussions in a purely theoretical paper such as this one.
> >
> > All in all, I am very positive after the discussion phase with the authors and am willing to increase my rating, provided all these comments are appropriately discussed in the reviser version of the manuscript.

---

> > > ### Author Response · Authors · 2025-08-06
> > >
> > > We sincerely thank the reviewer for the thoughtful follow-up and for clarifying the questions. We greatly value these comments and will incorporate them into the revised manuscript.
> > >
> > > ### On the complexity of interpolation/approximation
> > >
> > > Thank you for clarifying the question.  In our setting, combining our token-distinguishability result with the interpolation complexity analysis in [1] should yield a parameter complexity of $\mathcal{O}(d^2 N n)$, which is linear in $N$ but still depends on the sequence length $n$, as expected for general $N$  length-$n$ sequences.
> > >
> > > Measure-theoretic works indeed reduce this dependence by imposing additional regularity on the targets.  For example, for continuous-layer transformer, [2] shows that under an error tolerance $\varepsilon$, the interpolation complexity can be reduced to $\mathcal{O}(d N)$. However, their results only apply to the case where the target points are all Dirac measures. Moreover, Furuya, de Hoop, and Peyré [3] establish their result based on a Stone–Weierstrass argument, which does not provides complexity results. The key advantage of their results is the independence of sequence length, but it is based on continuity of the target map, which restricts the complexity of the targets as sequence length increases.
> > >
> > > Therefore, while measure-theoretic results elegantly handle infinite context length under additional assumptions, they do not imply complexity bounds that are both explicit and independent of $n$ for general sequence-to-sequence interpolation.
> > >
> > > ---
> > >
> > > ### On the relation between the measure-theoretic view and our framework
> > >
> > > We fully agree that the measure-theoretic perspective is a fundamental and valuable viewpoint, especially for probing the limits of transformers beyond fixed context length.
> > >
> > > However, the main goal of our paper is to provide a unified analysis of the expressive power of transformer-type architectures, including variants with sparse, low-rank, or other non-standard attention mechanisms. These mechanisms often introduce position-dependent or even layer-dependent actions between tokens (e.g., sliding-window, periodic, or random connections) that do not translate naturally into a measure-theoretic framework merely by restricting the support of a measure. For example, in the variants proposed in [4], there are connection modes that depend on the modulus of the position; for BigBird [5], it is the combination of global, local and random attention; and for LinFormer [6], a projection of the attention matrix onto a low-rank one is used. While these variants depends on different discrete information, it is not straightforward to lift them to a measure theoretic function uniformly and elegantly.
> > >
> > > For this reason, we chose the finite-length sequence formulation as the most suitable framework to present our results. Nevertheless, we fully agree that the measure-theoretic perspective is complementary, and we will explicitly discuss this connection, along with the reviewer’s valuable insights, in the revised version.
> > >
> > > ---
> > >
> > > References:
> > >
> > > [1] Ruiz-Balet, D., & Zuazua, E. (2023). Neural ODE control for classification, approximation, and transport. SIAM Review, 65(3), 735-773.
> > >
> > > [2] Geshkovski, B., Rigollet, P., & Ruiz-Balet, D. (2024). Measure-to-measure interpolation using Transformers. arXiv preprint arXiv:2411.04551.
> > >
> > > [3] Furuya, T., de Hoop, M. V., & Peyré, G. Transformers are Universal In-context Learners. In The Thirteenth International Conference on Learning Representations.
> > >
> > > [4] Child, R., Gray, S., Radford, A., & Sutskever, I. (2019). Generating long sequences with sparse transformers. arXiv preprint arXiv:1904.10509.
> > >
> > > [5] Zaheer, M., Guruganesh, G., Dubey, K. A., Ainslie, J., Alberti, C., Ontanon, S., ... & Ahmed, A. (2020). Big bird: Transformers for longer sequences. Advances in neural information processing systems, 33, 17283-17297.
> > >
> > > [6] Wang, S., Li, B. Z., Khabsa, M., Fang, H., & Ma, H. (2020). Linformer: Self-attention with linear complexity. arXiv preprint arXiv:2006.04768.

---

> > > > ### Comment · Reviewer_gCVj · 2025-08-06
> > > >
> > > > I thank the authors for their comments and clarifications. Adding these revisions are convincing for me. I'll increase my rating appropriately.

---

### Official Review · Reviewer_8Xaz · 2025-07-02

**Clarity:** 3
**Significance:** 3
**Originality:** 3
**Rating:** 4
**Confidence:** 1

**Summary:**

This paper presents a unified theoretical framework for analyzing the universal approximation property (UAP) of transformer-type architectures. Authors leverage token distinguishability as a fundamental condition for UAP to derive a general sufficient condition. Departing from previous work that focuses on architecture-specific constructive proofs, the authors introduce general, verifiable sufficient conditions that guarantee UAP across a broad class of transformer models with an attention-based module.

**Questions:**

No.

**Ethical Concerns:**

["NO or VERY MINOR ethics concerns only"]

**Final Justification:**

After reading the authors' and other reviewers' comments, I'll keep the same rating for weak acceptance, due to the broad applicability of the framework to various transformer architectures and mathematical rigor.

**Limitations:**

The authors did mention the limitation in the final chapter, and I don't think this work hasa  potential societal impact.

**Paper Formatting Concerns:**

No formatting concerns.

**Quality:**

3

**Strengths And Weaknesses:**

Strengths:
(1) The paper demonstrates the broad applicability of the framework to various transformer architectures, including: Kernel-based attention, Sparse attention, Linformer, Skyformer, etc.
(2) The paper is mathematically rigorous and provides extensive appendices with detailed proofs
(3) The framework recovers and generalizes prior UAP results on softmax-based transformers.

Weakness:
(1) Limitation for practical usage: Normalization layers, which are practically crucial for stable training, are omitted in the analysis.
(2) The work is purely theoretical and lacks any empirical experiments or validation. Some numerical results would help demonstrate practical implications.

---

> ### Author Rebuttal · Authors · 2025-07-29
>
> We sincerely thank the reviewer for taking the time to assess our work and for the constructive comments.
>
> ## On normalization layers and empirical validation
> - We acknowledge that normalization layers are crucial for practical transformer architectures. As noted in our response to Reviewer #1, extending the theoretical analysis to include normalization is a promising direction for future work.
>
> - Regarding the absence of empirical experiments, we emphasize that our contribution is theoretical in nature. Nevertheless, we agree that validating the practical implications of the framework through numerical experiments is an interesting future direction.

---

> > ### Comment · Reviewer_8Xaz · 2025-08-04
> > **Thank you for your response.**
> >
> > Thank you for your response. Authors response partially address my concerns and I'll keep my original score.

---

### Official Review · Reviewer_J3Ee · 2025-07-03

**Clarity:** 4
**Significance:** 2
**Originality:** 3
**Rating:** 5
**Confidence:** 3

**Summary:**

This paper studies the universal approximation/expressivity ability of transformer-type architectures from a theoretical perspective. The results concern the family of architectures that resembles a simplified transformer: attention with residual connection, followed by a MLP with residual connection. Using any such family of transformers, the authors prove a non-constructive expressivity result for the set of functions that satisfy a notion of permutation/symmetric invariance. The sufficient condition is affine invariance, and more importantly, token distinguishability for the attention-type function class.

**Questions:**

- Are $C_n$ and $D_n$ invariant functions realistic for most practical applications? My impression is that in practice, the order of tokens does matter, and it would be interesting to see if the framework can be extended to function classes that don't satisfy permutation equivariance.

**Ethical Concerns:**

["NO or VERY MINOR ethics concerns only"]

**Final Justification:**

As I stated in my response to the authors, I believe this work is worth a poster at the conference. The theory is sound, but I did not give it a higher score due to the slight insignificance of the work (e.g. see discussion by reviewer gCVj), as well as the unsurprising nature of the result.

**Limitations:**

Yes.

**Quality:**

3

**Strengths And Weaknesses:**

Strengths:
- The paper provides a clean set of sufficient conditions for which UAP is achieved for a given family of transformers, and is overall well written.
- A sufficient condition for the token distinguishability criterion is given that is feasible to check.
- Extensions to general transformers via kernels and sparse attention are provided too.
- Examples of different symmetry groups for the function class to be expressed are given, including the cyclic group and dihedral group.

Weaknesses:
- The results are not too surprising, since the moment you have token distinguishability, one would intuitively expect expressivity results to go through with enough symmetry.
- The architecture class does not include other components of the transformers architecture, such as normalization.

---

> ### Author Rebuttal · Authors · 2025-07-29
>
> We sincerely thank the reviewer for the positive and insightful feedback.
>
> ## On the lack of normalization layers
> We acknowledge the omission of normalization layers as a limitation. This is a common simplification in theoretical analyses of transformer architectures, intended to isolate core components such as attention and feedforward layers. From a mathematical perspective, normalization introduces a nonlinear projection onto a product of unit spheres, altering the function family $\mathcal{T}_{\mathcal{G}, \mathcal{H}}$. Inspired by recent theoretical studies on dynamics constrained to spheres (e.g., [1]), we believe that extending our framework to include normalization is a promising future direction.
>
> ## On $D_n$ and $C_n$ symmetry
> Our intention is not to claim that $D_n$ or $C_n$ symmetries are typical in practical tasks. Rather, these examples serve to illustrate how our framework enables principled design of transformer architectures respecting general symmetry constraints—beyond the full permutation group. Such symmetries do arise in specific scientific domains: $D_n$ symmetry corresponds to the structure of regular molecules, while $C_n$ symmetry is relevant for periodic data, as we have mentioned in line 339-341. Other group symmetries (e.g., alternating groups, space groups) also appear in physics and chemistry, and could be explored
>
> ## On functions without symmetry
> Our framework naturally encompasses functions without symmetry as the special case where the symmetry group $G = { \text{Id} }$. Also, as noted in lines 119–122, prior work shows that for any architecture family satisfying $G$-UAP with nontrivial $G$, adding a suitable positional encoding allows approximation of general, non-equivariant functions. Hence, our focus on $G$-UAP is not restrictive, but rather provides a unified treatment of both symmetric and non-symmetric function classes.
>
> Reference:
> [1] Geshkovski, B., Rigollet, P., & Ruiz-Balet, D. (2024). Measure-to-measure interpolation using Transformers. arXiv:2411.04551.

---

> ### Comment · Reviewer_J3Ee · 2025-08-03
>
> I thank the authors for their rebuttal, and I believe this would be worth a poster at the conference. I will keep my original rating.

---

### Official Review · Reviewer_rM6S · 2025-07-03

**Clarity:** 3
**Significance:** 3
**Originality:** 3
**Rating:** 5
**Confidence:** 3

**Summary:**

This manuscript investigates the expressivity of transformer-like architectures with various attention mechanisms, and establishes their universal approximation property (UAP).  It does so via token-sequence distinguishability under the action of a symmetric subgroup G, followed by sufficiently rich FFNs.  The authors demonstrate the strength of their results via application to various attention mechanisms, and by constructing new architectures equipped with UAP by design.

**Questions:**

I feel like there should be a more intuitive way to write Eq. (4), which is after all a pretty straightforward parallel application of $\mathcal{H}$ at each position.  I nudge the authors to consider an alternative, but it’s not crucial.

Nitpick:
Below Eq. (1) the authors repeatedly use “token” to mean something like “the vector in the residual stream at some layer and token position”.  Typically I think of tokens as elements of a finite set.  The embeddings in the residual stream are then no longer “tokens”.  Alas, my substitution would be unwieldy, but perhaps a brief comment on terminology could be added.  For example, the authors could add a footnote after Eq. (1) saying something like “We will abuse language slightly and call any $d$-dimensional column of $X_t$​ a ‘token’, even though it is the continuous embedding of a discrete input symbol.”

Minor suggestions:

Line 102:
“can but only can represent only” appears to be a typo.  The authors may want to replace this with something like “can represent only”

Line 102:
Would “tensor-product functions” be more accurate than “tensor-type functions”?

**Ethical Concerns:**

["NO or VERY MINOR ethics concerns only"]

**Final Justification:**

I remain satisfied with the strengths of this paper and am happy to maintain my score and recommend it for a poster.

**Limitations:**

yes

**Paper Formatting Concerns:**

no such concerns

**Quality:**

4

**Strengths And Weaknesses:**

This paper appears novel in that it establishes UAP in a relatively architecture-independent fashion for transformer-type models.

The paper is very well written.  It builds up the notation and ideas intuitively and precisely, has an elegant aesthetic, and guides the reader easily to and through the results.

UAP has been established for standard transformer architectures.  The surface-level role of this paper is to extend these results to variations of that architecture, which feels somewhat iterative.  However, the techniques developed in this paper seem to provide useful broad insights, and will likely serve more utility beyond the specific results derived here.

---

> ### Author Rebuttal · Authors · 2025-07-29
>
> We sincerely thank the reviewer for your positive and insightful feedback.
>
> ## On notation and minor suggestions
>
> - We acknowledge that the use of the term "token" may be potentially ambiguous. Following your suggestion, we will add a footnote after Eq. (1) clarifying that we slightly abuse terminology by referring to any column of $X_t$ as a "token", even though it is the continuous embedding of a discrete input symbol.
>
> - We appreciate your careful reading and will correct the typo in line 102. We will also adopt your suggestion to replace "tensor-type functions" with "tensor-product functions" for better clarity.

---

> > ### Comment · Reviewer_rM6S · 2025-08-06
> > **Reply to authors' response**
> >
> > I remain satisfied with the strengths of this paper and am happy to maintain my positive recommendation.

---

### Decision · Program_Chairs · 2025-09-17

**Decision:**

Accept (poster)

**Comment:**

This is a nice theoretical paper that studies the universal approximation property of Transformers as well as variants of Transformers. Reviewers unanimously voted in favor of accept, and the AC agrees that the paper adds value to understanding and formalizing Transformers.